# Mega-scale experimental analysis of protein folding stability in biology and design

Kotaro Tsuboyama[1,2,3,11], Justas Dauparas[4,5], Jonathan Chen[1,2,6], Elodie Laine[7], Yasser Mohseni Behbahani[7], Jonathan J. Weinstein[8], Niall M. Mangan[2,9], Sergey Ovchinnikov[10] & Gabriel J. Rocklin[1,2✉]

Advances in DNA sequencing and machine learning are providing insights into protein sequences and structures on an enormous scale[1]. However, the energetics driving folding are invisible in these structures and remain largely unknown[2]. The hidden thermodynamics of folding can drive disease[3,4], shape protein evolution[5–7] and guide protein engineering[8–10], and new approaches are needed to reveal these thermodynamics for every sequence and structure. Here we present cDNA display proteolysis, a method for measuring thermodynamic folding stability for up to 900,000 protein domains in a one-week experiment. From 1.8 million measurements in total, we curated a set of around 776,000 high-quality folding stabilities covering all single amino acid variants and selected double mutants of 331 natural and 148 de novo designed protein domains 40–72 amino acids in length. Using this extensive dataset, we quantified (1) environmental factors influencing amino acid fitness, (2) thermodynamic couplings (including unexpected interactions) between protein sites, and (3) the global divergence between evolutionary amino acid usage and protein folding stability. We also examined how our approach could identify stability determinants in designed proteins and evaluate design methods. The cDNA display proteolysis method is fast, accurate and uniquely scalable, and promises to reveal the quantitative rules for how amino acid sequences encode folding stability.

Protein sequences vary by more than ten orders of magnitude in thermodynamic folding stability[2] (the ratio of unfolded to folded molecules at equilibrium). Even single point mutations that alter stability can have profound effects on health and disease[3,4], pharmaceutical development[8–10] and protein evolution[5–7]. Thousands of point mutants have been individually studied over decades to quantify the determinants of stability[11], but these studies highlight a challenge: similar mutations can have widely varying effects in different protein contexts, and these subtleties remain difficult to predict despite substantial effort[12,13]. In fact, even as deep learning models have achieved transformative accuracy at protein structure prediction[1], progress in modelling folding stability has arguably stalled[14,15]. New high-throughput experiments have the potential to transform our understanding of stability by quantifying the effects of mutations across a vast number of protein contexts, revealing new biophysical insights and empowering modern machine learning methods.

Here we introduce cDNA display proteolysis, a powerful high-throughput stability assay, and use it to produce a large dataset of 776,298 folding stability measurements. This method combines the strengths of cell-free molecular biology and next-generation sequencing and requires no on-site equipment larger than a quantitative PCR

(qPCR) instrument. Assaying one library (up to 900,000 sequences in our experiments) requires one week and reagents costing about US$2,000, excluding the cost of DNA synthesis and sequencing. Compared with mass spectrometry-based high-throughput stability assays[16,17], cDNA display proteolysis achieves a 100-fold larger scale and can easily be applied to study mutational libraries that pose difficulties for proteomics. Compared with the previous yeast display proteolysis method[18], cDNA display proteolysis resolves a wider dynamic range of stability and is more reproducible even at a 50-fold larger experimental scale. Large-scale proteolysis data have already had a key role in the development of machine learning methods for protein design and protein biophysics[19,20]. The cDNA display proteolysis method massively expands this capability and has the potential to expand our knowledge of stability to the scale of all known small domains.

The dataset of 776,298 absolute folding stabilities is unique in size and character. Current thermodynamic databases contain a skewed assortment of mutations measured under many varied conditions[11]. By contrast, this new dataset comprehensively measures all single mutants for 331 natural domains and 148 designed proteins—including single deletions and two insertions at each position—all under identical conditions. Our dataset also includes comprehensive double mutations

[1]Department of Pharmacology, Northwestern University Feinberg School of Medicine, Chicago, IL, USA. [2]Center for Synthetic Biology, Northwestern University, Evanston, IL, USA. [3]PRESTO, Japan Science and Technology Agency, Tokyo, Japan. [4]Department of Biochemistry, University of Washington, Seattle, WA, USA. [5]Institute for Protein Design, University of Washington, Seattle, WA, USA. [6]McCormick School of Engineering, Northwestern University, Evanston, IL, USA. [7]Sorbonne Université, CNRS, IBPS, Laboratory of Computational and Quantitative Biology (LCQB), UMR 7238, Paris, France. [8]Department of Biomolecular Sciences, Weizmann Institute of Science, Rehovot, Israel. [9]Department of Engineering Sciences and Applied Mathematics, Northwestern University, Evanston, IL, USA. [10]John Harvard Distinguished Science Fellowship Program, Harvard University, Cambridge, MA, USA. [11]Present address: Institute of Industrial Science, The University of Tokyo, Tokyo, Japan. ✉e-mail: grocklin@gmail.com

at 559 site pairs spread across 190 domains (a total of 210,118 double mutants). We used this unique dataset to investigate how individual amino acids and pairs of amino acids contribute to folding stability as well as how selection for stability interacts with other selective pressures in natural protein domains. We also explored how our unique scale of data can be applied in protein design.

## The cDNA display proteolysis method

Proteases typically cleave unfolded proteins more quickly than folded ones, and proteolysis assays have long been used to measure folding stability[21] and select for proteins with high stability[22,23]. We developed cDNA display proteolysis to efficiently measure folding stability using next-generation sequencing, following similar principles to the yeast display proteolysis method[18]. Each experiment begins with a DNA library. We used synthetic DNA oligonucleotide pools in which each oligonucleotide encodes one test protein. The DNA library is transcribed and translated using cell-free cDNA display[24], which is based on mRNA display[25,26], resulting in proteins that are covalently attached to their cDNA at the C terminus. We then incubate the protein–cDNA complexes with different concentrations of protease, quench the reactions, and pull down the proteins using an N-terminal PA tag (Fig. 1a). Intact (protease-resistant) proteins remain attached to their C-terminal cDNA. We then determine the relative amounts of all proteins in the surviving pool at each protease concentration by deep sequencing (Fig. 1c). To control for effects of protease specificity, we perform separate experiments with two orthogonal proteases: trypsin (targeting basic amino acids) and chymotrypsin (targeting aromatic amino acids).

We inferred the protease stability of all sequences from our sequencing counts using a Bayesian model of the experimental procedure. We modelled protease cleavage using single turnover kinetics[27,28] (Fig. 1b, equations (1) to (3)) because we assume that the enzyme is in excess over all substrates (up to about 20 pM of substrate[24] versus at least 141 pM of protease). To parameterize the model, we used a universal $k_{max}$ (maximum cleavage rate) for all sequences (Extended Data Fig. 1) and used our sequencing data to infer a unique $K_{50}$ (the protease concentration at which the cleavage rate is half of $k_{max}$; Fig. 1d and Methods) for each sequence. The inferred $K_{50}$ values were consistent between 2 replicates of the proteolysis procedure ($R = 0.97$ for trypsin and 0.99 for chymotrypsin for around 84% of sequences in a pool of 806,640 sequences after filtering based on confidence and dynamic range).

To infer each sequence's thermodynamic folding stability (for unfolding at pH 7.4 and 298 K (room temperature); hereafter referred to as $\Delta G$), we used a kinetic model that separately considers idealized folded (F) and unfolded (U) states (Fig. 1b, equation (4)). We model both states using the same single-turnover equations as before (Fig. 1b, equation (3)) with separate $K_{50}$ protease concentrations for each state ($K_{50,F}$ and $K_{50,U}$) and a universal, shared $k_{max}$. We assume that cleavage in the folded state occurs exclusively in the constant regions of the construct such as the N-terminal PA tag, so we use an identical $K_{50,F}$ for all sequences. By contrast, $K_{50,U}$ reflects an individual sequence's unique protease susceptibility in the unfolded state, which depends on its potential cleavage sites (Fig. 1e). We inferred $K_{50,U}$ for each sequence using a position-specific scoring matrix model parameterized using measurements from 64,238 scrambled sequences that are likely to be fully unfolded (Extended Data Fig. 2 and Methods). Inferring a unique $K_{50,U}$ for each sequence accounts for differences in unfolded state susceptibility between sequences, but any cleavage from folded or partially-folded states cannot be corrected by this model. Finally, we assume that folding, unfolding and enzyme binding are all in rapid equilibrium relative to cleavage, implying that $K_{50,U}$, $K_{50,F}$ and the overall $K_{50}$ can be approximated by the enzyme–substrate equilibrium dissociation constants (Fig. 1b, equation (6)). Although these approximations will not be universally accurate, they are often valid for small domains and facilitate consistent analysis of all sequences. With these approximations, we

can determine the $\Delta G$ for a sequence from its experimentally measured $K_{50}$, its inferred $K_{50,U}$, and the universal $K_{50,F}$ (Fig. 1b, equations (5) and (7) and Fig. 1e; derivation in Supplementary information).

Our model has notable limitations. First, stability ($\Delta G$) will be underestimated if significant cleavage occurs inside the test domain from folded or partially folded states (that is, without global unfolding). Second, stability can be over- or under-estimated depending on the accuracy of $K_{50,U}$ (independent measurements with trypsin and chymotrypsin help correct this). Third, $\Delta G$ values become unreliable if $K_{50}$ approaches $K_{50,F}$ or $K_{50,U}$ (Fig. 1e). Owing to these limitations, we developed data quality filters to remove unreliable $\Delta G$ estimates (see below). After filtering, the $\Delta G$ values inferred by the model were consistent between our independent experiments with trypsin and chymotrypsin ($R = 0.94$; Fig. 1f). For most analyses, we combined trypsin and chymotrypsin data into a single overall $\Delta G$ estimate (Methods).

## High-throughput data are accurate

Our cDNA display proteolysis measurements are highly consistent with published studies using purified protein samples for 1,188 variants of 10 proteins (Fig. 1g and Supplementary Fig. 1 for more details on GB1[29]). All Pearson correlations are above 0.75. Our measurements for these sequences were all performed in libraries of 244,000–900,000 total sequences. Several sets of mutants show systematic offsets ($y$-intercept values) between literature values and our measurements. We attribute these offsets to temperature differences between our conditions and the published experiments, and the offsets are correlated with temperature (except for NTL9 (Protein Data Bank (PDB) ID: 2HBB); Extended Data Fig. 3; see Supplementary Table 1 for all experimental conditions and references). The consistency between our cDNA display proteolysis results and traditional experiments establishes that (1) small domains like these are cleaved mainly in the globally unfolded state, (2) our method can reliably measure these cleavage rates on a massive scale, and (3) our unfolded state model can remove protease-specific effects to infer accurate quantitative folding stabilities.

## Mutational scanning of diverse domains

To systematically examine how individual residues influence folding stability, we used cDNA display proteolysis to measure stability for all single substitutions, deletions and Gly and Ala insertions in 983 natural and designed domains (wild-type sequences). We selected our natural domains to cover nearly all the small (less than 72 amino acids) monomeric domains in the PDB that were suitable for our assay (Methods). To minimize any cleavage from the folded state, we used AlphaFold models of each domain to remove unstructured terminal segments from each sequence. Our designed domains included (1) previous Rosetta designs with ααα, αββα, βαββ, and ββαββ topologies[18,30] (40 to 43 amino acids), (2) new ββαα proteins designed using Rosetta (47 amino acids), and (3) new domains designed by trRosetta hallucination[31,32] (46 to 69 amino acids). Note that the structures of these designs have not been validated experimentally. Our 983 wild-type sequences also include 121 'destabilized wild-type backgrounds' designed to resolve the effects of mutants on highly stable domains (Extended Data Fig. 4). We collected these data using four giant synthetic DNA oligonucleotide libraries and obtained $K_{50}$ values for 2,520,337 sequences; 1,841,285 of these measurements are included here. $K_{50}$ values were reproducible across libraries (Extended Data Fig. 5).

Deep mutational scanning of hundreds of domains revealed several overall patterns. The largest fraction of these domains showed clear, biophysically reasonable sequence–stability relationships that were consistent between independent experiments with trypsin and chymotrypsin. However, other domains were completely unfolded, too stable to resolve or produced inconsistent results between the proteases (Fig. 2b). To construct a reliable dataset of $\Delta G$ and $\Delta\Delta G$ measurements

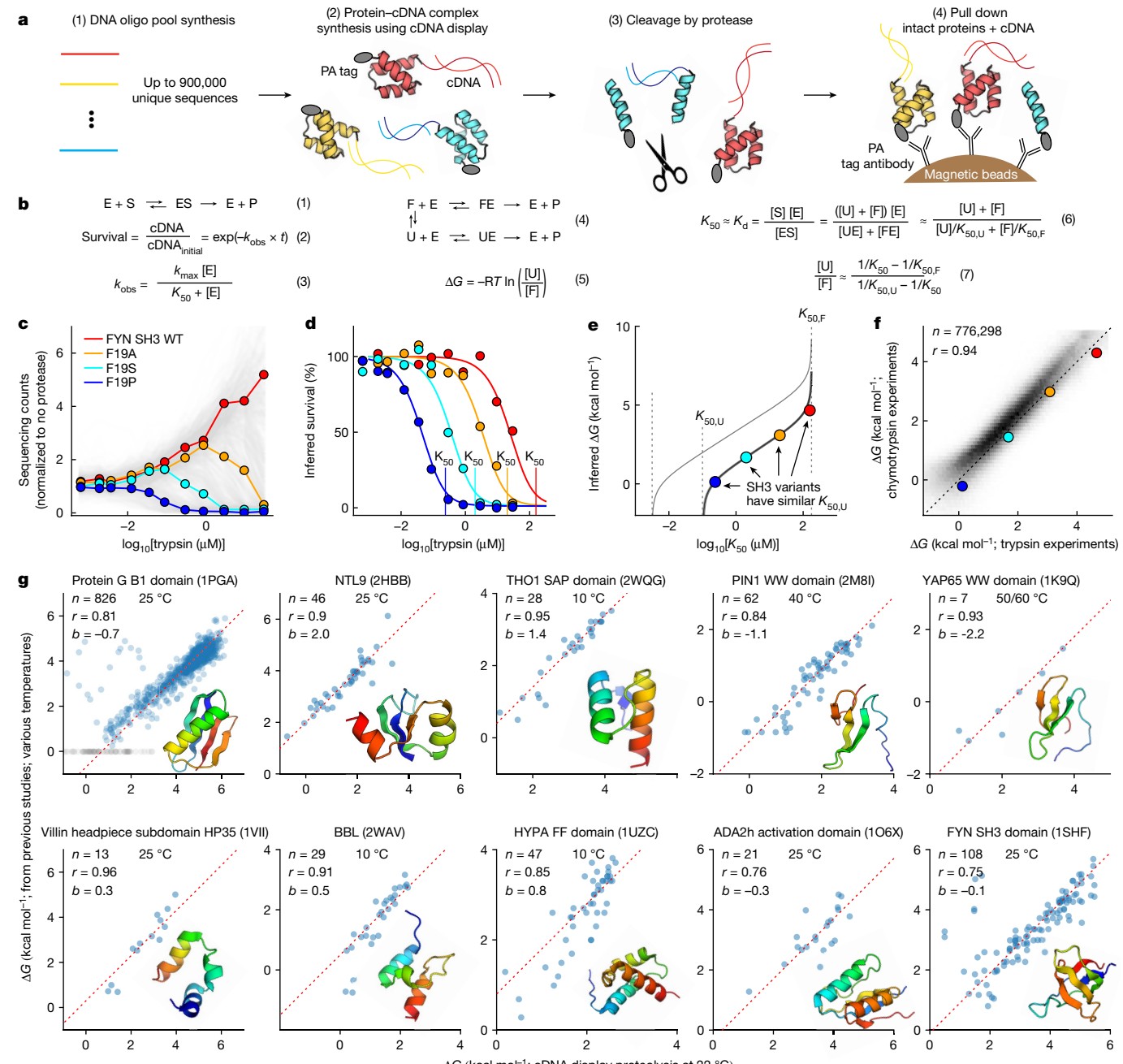

**Fig. 1 | cDNA display enables massively parallel measurement of protein folding stability. a**, A DNA oligonucleotide (oligo) library is expressed using cell-free cDNA display, producing proteins with an N-terminal 14-amino-acid PA tag and C-terminal covalent linkage to cDNA. After protease challenge, magnetic beads with anti-PA antibodies pull down proteins by their N termini. Sequencing the cDNA that is pulled down with the intact proteins enables quantification of the distribution of intact proteins. **b**, A thermodynamic model of proteolysis. (1) Protease enzymes (E) and protein substrates (S) form an ES complex to produce cleaved protein products (P). We model the cleavage as a first-order reaction (2) according to single-turnover kinetics (3). (4) Proteins are normally cleaved in the unfolded (U) state but can also be cleaved in the folded (F) state by cleaving the PA tag. We determine $\Delta G$ using each sequence's measured $K_{50}$, a predicted sequence-specific $K_{50}$ for the unfolded state ($K_{50,U}$), and a universal $K_{50}$ for the folded state ($K_{50,F}$) (5–7). $K_d$, dissociation constant. $k_{obs}$, observed rate constant. **c**, Each human FYN SH3 variant sequence is shown as a grey line tracking its sequencing counts (fraction of the total library) relative to the

pre-selection library. Four variants are highlighted in colour. WT, wild type. **d**, Inferred survival of uncleaved protein for four sequences from **c** at different protease concentrations (dots); lines show the global fit from the kinetic model. Vertical lines show inferred $K_{50}$ concentrations (one-half maximal cleavage rate, not 50% total cleavage). **e**, Relationship between $K_{50}$ and $\Delta G$ for different values of $K_{50,U}$. SH3 variants (coloured circles) all have similar $K_{50,U}$ and fall on nearly the same $\Delta G$ versus $K_{50}$ line (black). Sequences with more cut sites have lower $K_{50,U}$ and higher $\Delta G$ estimates for any $K_{50}$ (grey line). **f**, Consistency of $\Delta G$ estimates between independent trypsin and chymotrypsin experiments after quality filters (dataset 2), highlighting proteins shown in **c**. **g**, Our high-throughput $\Delta G$ measurements agree with published data from purified samples for mutants of the indicated domains (PDB IDs are in parentheses). Dashed lines show $y = x + b$ (intercept). Grey points indicate missing reference data for the B1 domain of protein G[29]. Plots indicate number of points ($n$), Pearson correlation ($r$), $y$-intercept ($b$) and the temperatures used for purified protein experiments (Supplementary Table 1). Insets show structural models.

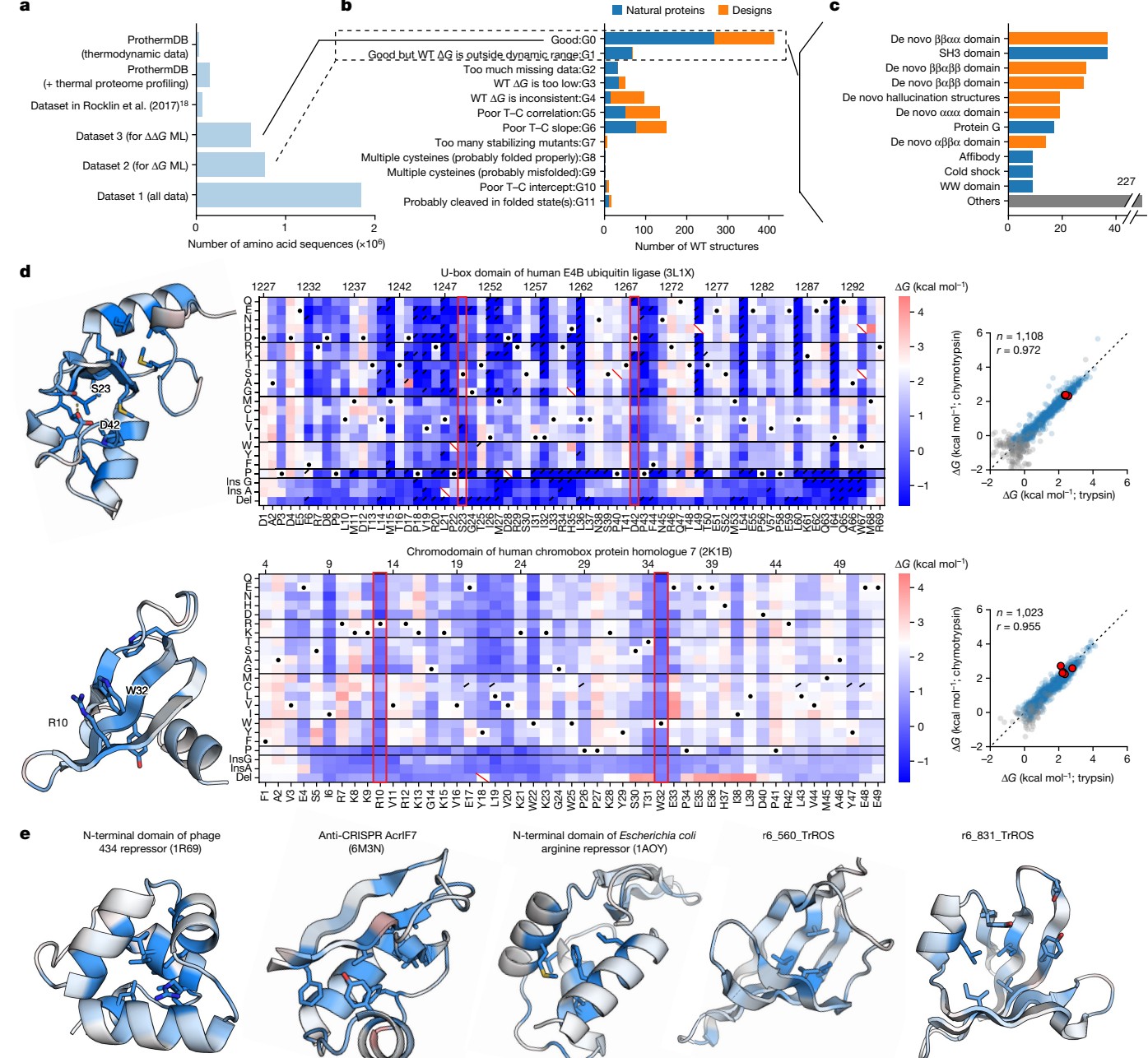

**Fig. 2 | Comprehensive mutational analysis of stability in designed and natural proteins. a**, The size of existing datasets compared with our datasets 1, 2 and 3. Datasets 1, 2 and 3 are defined in Extended Data Table 1. ML, machine learning. **b**, Classification of mutational scanning results for each wild-type (WT) sequence. Each wild type sequence is included in only one group even when that wild-type meets multiple criteria (for example, both a poor slope and inconsistent intercept between trypsin and chymotrypsin (T–C)). **c**, Wild-type structures classified as G0 in **b** grouped into domain families. The 11 most common domain types are shown; additional classification shown in Supplementary Fig. 3. **d**, Mutational scanning results for the U-box domain of human E4B ubiquitin ligase (PDB ID: 3L1X) (top) and chromo domain of the chromobox protein homologue 7 (PDB ID: 2K1B) (bottom). Left, domain structures coloured by the average ΔΔ$G$ at each position; darker blue indicates that mutants are more destabilizing. Middle, heat maps show Δ$G$ for substitutions, deletions and Gly and Ala insertions at each residue, with PDB numbering at top and our one-indexed numbering at bottom. White indicates wild-type stability, and red and blue indicate stabilizing and destabilizing mutations, respectively. Black dots indicate wild-type amino acids, red slashes indicate missing data and corner slashes indicate lower confidence Δ$G$ estimates (95% confidence interval > 0.5 kcal mol⁻¹), including Δ$G$ estimates near the edges of the dynamic range. Red boxes highlight the S23–D42 hydrogen bond in the U-box domain of human E4B ubiquitin ligase and the R10–W32 cation–π interaction in the chromo domain of the chromobox protein homologue 7. Δ$G$ values were fitted to trypsin and chymotrypsin data together (Methods). Right, Δ$G$ values independently determined using assays with trypsin ($x$ axis) and chymotrypsin ($y$ axis). Multiple codon variants of the wild-type sequence are shown in red, reliable Δ$G$ values are in blue, and less reliable Δ$G$ estimates are in grey. The black dashed line shows $y = x$. Plots show the number of reliable points and the Pearson $r$ value for the blue (reliable) points. **e**, Structures of five other domains in our datasets, presented as in **d**. The two designed structures are AlphaFold models.

from folded domains, we filtered the mutational scans on the basis of the consistency of the trypsin and chymotrypsin $\Delta G$ estimates and other criteria designed to remove domains showing cleavage from folded or partially folded states (Supplementary Fig. 2 and Methods). Many domains that showed evidence of cleavage from folded states had cleavable sites near the termini or in flexible loop regions. Indeed, four of our deep mutational scans use modified wild-type backgrounds in which folded state cleavage sites (identified in earlier mutational scans) had been removed (Extended Data Fig. 6a–c). However, other domains with long loops appeared fully cooperative, without evidence of cleavage of the folded state (Extended Data Fig. 6d). Along with susceptibility to cleavage of folded or partially-folded states, other domains taken from the PDB failed in our assay for a variety of reasons, including poor cell-free expression (category G2), structures that were probably stabilized by crystal contacts (PDB ID: 2CUW and 2FGG), missing N–C cyclization incompatible with cDNA display (PDB ID: 1E68 and 2MP8), poor stability at room temperature (PDB ID: 2C0S and 2LGN), and large numbers of cysteine residues (category G9). Our final quality-filtered datasets are shown in Extended Data Table 1 and Supplementary Fig. 3.

Mutational scanning results are shown for seven domains in Fig. 2d,e. Like all mutational scans that passed filtering, these examples show a strong consistency between independent $\Delta G$ measurements with trypsin and chymotrypsin (Pearson correlation 0.94 ± 0.04 (median ± s.d.) for 478 domains in dataset 2). The most critical sites (those where mutations are highly destabilizing) are located in the hydrophobic core. However, our data also reveal many other critical interactions, such as a side chain hydrogen bond between S23 and D42 in the U-box domain of human E4B ubiquitin ligase and a cation–π interaction between R10 and W32 in the chromodomain of human chromobox protein homologue 7 (residues have been renumbered based on the exact sequence included in our experiments). These unique stabilizing interactions reveal the rich biophysical diversity found in our systematic exploration of stability across hundreds of domains.

## Global trends in amino acid fitness

We first sought to define the major sources of variation between protein sites that influence the relative stabilities of all 20 amino acids at that site. To this end, we performed principal component analysis using 325,132 $\Delta G$ measurements at 17,093 sites in 365 domains (dataset 3). Each principal component expresses specific properties of a site that determine which amino acids are stabilizing or destabilizing. Based on the loadings of the amino acids onto each principal component (Fig. 3a), we interpreted the first four components to reflect whether a site is stabilized by hydrophobic amino acids (principal component 1 (PC1); 31% of the total variance explained by this principal component), helix-favouring amino acids (PC2; 15%), aliphatic versus aromatic amino acids (PC3; 12%), and positive versus negative charges (PC4; 7%). The fifth principal component (6%) was more complex: at one extreme were small amino acids that could be buried in dense environments, along with basic amino acids that can 'snorkel' their charged moieties to the surface even when partially buried. At the other extreme were acidic amino acids that are energetically costly to bury. We interpreted this component to reflect volumetric properties of buried sites that are orthogonal to the properties captured by PC1. These interpretations are also consistent with the structural environments at each site (Fig. 3b). Figure 3c illustrates the first five principal components at all sites in the C-terminal domain of the transcription factor NusG. Sites with positive principal component values (pink) are stabilized by amino acids with positive loadings on that component (Fig. 3a). Visualizing these principal components on this NusG domain highlights the hydrophobic core (PC1), differing regions of the core that favour aromatic or aliphatic residues (PC3), sites stabilized by acidic or basic residues (PC4) and a critical location for a small amino acid (PC5). These structural

characteristics (and the meanings of the principal components) can also be seen by directly comparing the principal component values to the mutational scanning stability data (Fig. 3d).

Whereas the first principal components capture the main sources of variation between sites, the other components capture more subtle factors that still influence stability. We found that these later principal components capture a greater amount of the total variance for natural protein domains compared to designed proteins, although the difference between natural domains and Rosetta designs is slight (Fig. 3e). This indicates that the amino acid environments in natural domains are on average more complex than those in our set of designed proteins; more subtle contributions to stability have a larger role. Sites in natural domains also have greater total variance in stability between different amino acids (Fig. 3f), although we note that our natural domains are larger on average than our designed domains: 57 ± 10 amino acids versus 44 ± 3 amino acids. Finally, we found that the subtle effects captured by the later principal components still stabilize wild-type amino acids. Across all sites, the median $\Delta\Delta G$ is −0.59 kcal mol$^{-1}$, indicating that the wild type is typically more stable than an alternative amino acid. However, the first five principal components can only explain 0.36 kcal mol$^{-1}$ (approximately 61%) of this stability difference; the remainder is attributable to the other components (Extended Data Fig. 7a). This indicates that the remaining components capture additional biophysical effects that contribute to the compatibility between wild-type amino acids and their environments, especially for natural domains (Extended Data Fig. 7b).

## Large-scale thermodynamic coupling data

Next, we examined how side chain interactions between amino acid pairs affect stability. We constructed comprehensive substitutions (20 × 20 amino acids) of 559 amino acid pairs from 190 natural domains and designs and measured stability for all sequences by cDNA display proteolysis. We selected pairs that were suggested to form energetically important hydrogen bonds in our mutational scanning data as well as other pairs forming close contacts (Fig. 4a; Methods). To quantify the interactions between side chains, we built an additive model for each amino acid pair with 40 parameters that capture the independent stability contributions of each amino acid in each position. The deviations from these models quantify the 'thermodynamic coupling' between specific amino acids[33]. Among our curated set of wild-type pairs, couplings were typically 0.5–1.0 kcal mol$^{-1}$ in magnitude, with some greater than 2 kcal mol$^{-1}$ (Fig. 4b). Among all sequences tested (wild-type or mutant pairs), pairs with opposite charges and cysteine pairs tended to have positive (favourable) couplings, whereas pairs with the same charge and acidic-aromatic–aliphatic pairs tended to have negative couplings (Fig. 4c). Average couplings are lower than wild-type couplings because the side chain orientations and environment surrounding wild-type pairs will typically be optimized for that pair. Still, our data recapitulate expected patterns of side chain interactions, provide a wealth of data for training machine learning models, and identify a wide range of noteworthy interactions for further study.

Several notable pairs are highlighted in Fig. 4d–f. In an OB-fold domain from *Shewanella oneidensis*, we found strong thermodynamic coupling between two unrelated pairs of amino acids: the wild-type Tyr–Glu pair and a mutant Lys–Trp pair that may form a cation–π interaction (thermodynamic couplings of 1.6 ± 0.2 and 1.4 ± 0.2 kcal mol$^{-1}$, respectively (mean ± s.d. from calculating the coupling using bootstrap resampling of the around 400 amino acid combinations); Fig. 4d and Supplementary Fig. 4a). In the α-spectrin SH3 domain, our comprehensive double mutant scanning of Y10 and Y52 uncovered the highly stable, tightly coupled double mutant Y10H/Y52K (coupling of 2.5 ± 0.4 kcal mol$^{-1}$ for His–Lys versus 1.0 ± 0.2 kcal mol$^{-1}$ for the wild-type pair; Fig. 4e and Supplementary Fig. 4b). AlphaFold modelling predicts that this double mutant introduces a new hydrogen

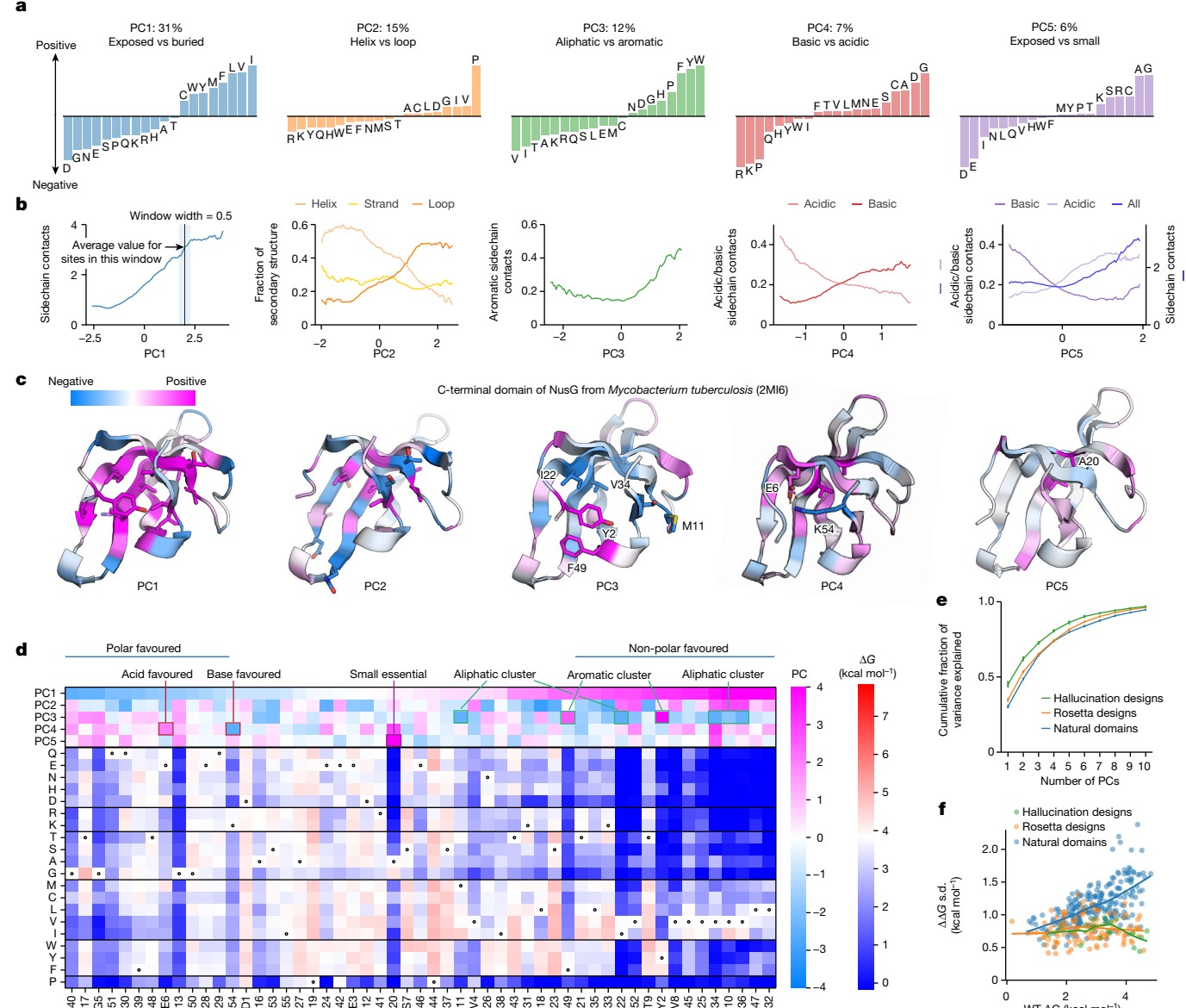

**Fig. 3 | Environmental factors that determine amino acid stabilities at a position. a**, Principal components (PC1–PC5) of the stability data showing the dominant trends for which amino acids are stabilizing or destabilizing at a site. We label each component with a biophysical interpretation and show the percentage of total variance explained by that component. Principal component analysis was performed using 20 features (ΔG with each amino acid variant) and 17,093 observations (sites in 365 domains). 325,132 ΔG measurements total. **b**, Relationships between the principal component values (*x* axis) for all 17,093 sites and environmental properties of those sites from the modelled 3D structure (*y* axis). Coloured lines show each environmental feature averaged over a window of 0.5 principal component units. Contacts indicate contact counts at those sites (Methods), counting all possible contacts (PC1 and PC5), only contacts to aromatic or aliphatic amino acids (PC3) or only contacts to acidic or basic amino acids (PC4). **c**, Structures of the C-terminal domain of NusG (PDB ID: 2MI6) with sites coloured by the value of PC1–PC5. Positive sites are

stabilized by amino acids with positive loadings on that principal component (**a**). **d**, Mutational scanning results for NusG ordered by the value of PC1 at each site (not sequence order). The first five principal components at each site are shown at the top, coloured blue (negative) to pink (positive). The stability effects of each substitution are shown at the bottom, coloured blue (destabilizing) to red (stabilizing). Several positions with large (positive or negative) principal component values are highlighted to show the correspondence between principal component values and specific mutational stability patterns from **c**. **e**, Cumulative fraction of the total variance explained by the principal components, for separate principal component analyses of natural domains (blue), Rosetta designs (orange) and hallucination designs (green). Error bars show the s.d. from bootstrap resampling (*n* = 1,000) of the mutational scans but are too small to observe. **f**, Relationship between wild-type stability and variance (s.d.) in the ΔΔG data at each site, coloured as in **e**. Lines show LOWESS fits.

bonding network to replace the original Tyr–Tyr interaction. We also identified an unexpected thermodynamic coupling between an amino acid pair lacking a direct side chain interaction. In the SH3 domain of Myo3, mutations at K24 are destabilizing even though the side chain makes no clear interactions. To investigate interactions of K24, we quantified thermodynamic couplings to nearby Y9 (0.0 ± 0.1 kcal mol⁻¹) and D10 (1.0 ± 0.2 kcal mol⁻¹) (Fig. 4f and Supplementary Fig. 4c). The

unexpected K24–D10 coupling—between two side chains that appear to not interact—highlights the difficulty of inferring energetic interactions from structural data alone and suggests a possible longer-ranged ionic interaction.

We also investigated thermodynamic couplings within 36 different 3-residue networks. For each triplet, we measured stability for all possible single and double substitutions in both the wild-type background

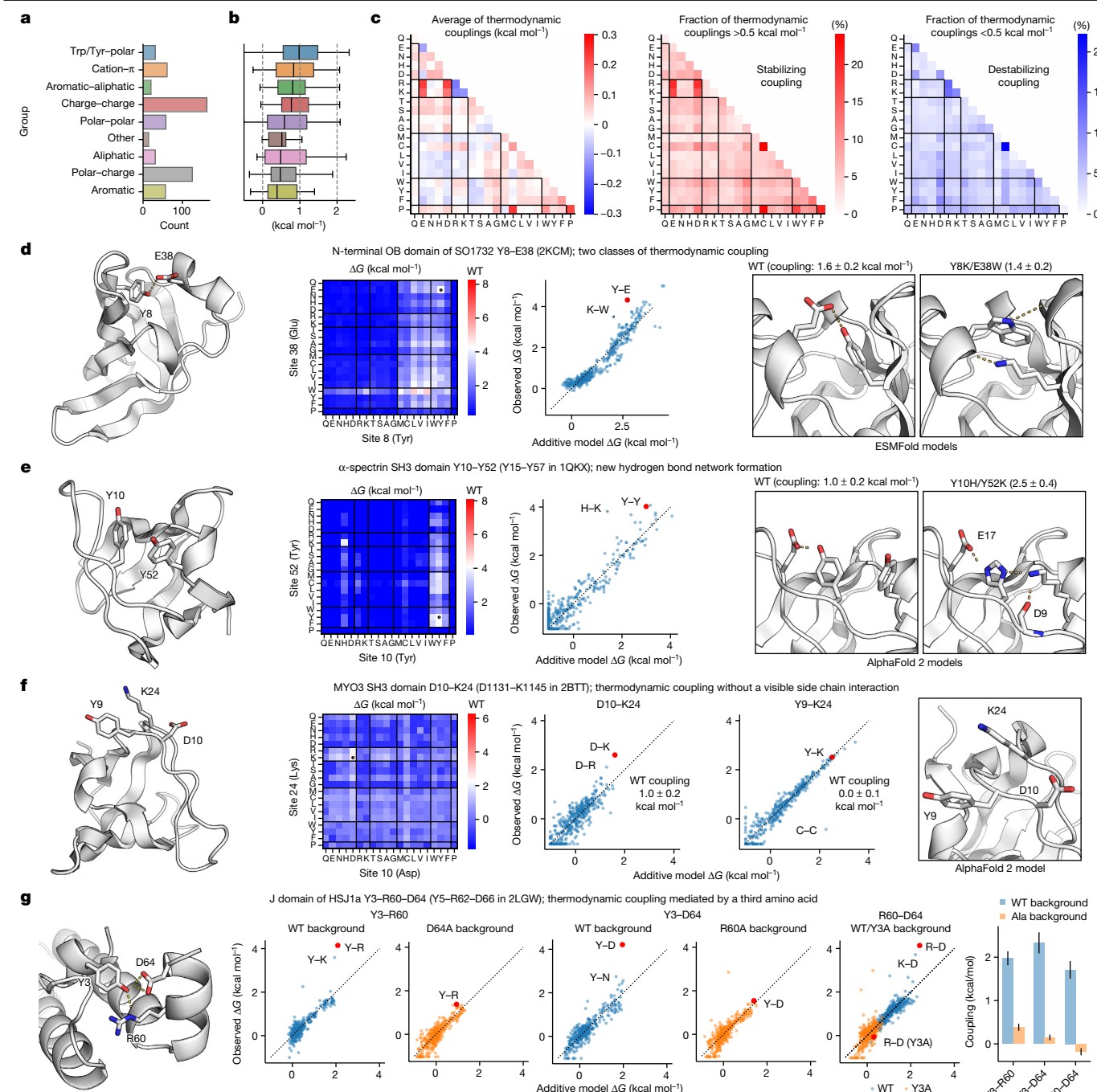

**Fig. 4 | Quantification of thermodynamic coupling between amino acid pairs. a,b**, Categorization of 559 pairs of amino acids in 190 domains selected for exhaustive double mutant analysis (210,000 $\Delta G$ measurements in total) (**a**) and thermodynamic couplings of wild-type amino acid pairs according to our additive model (**b**). In box plots, the centre line is the median, box limits delineate top and bottom quartiles and whiskers represent 1.5× the interquartile range ($n = 559$ in total). **c**, Average thermodynamic couplings (left) and fraction of pairs with positive (middle) or negative (right) couplings stronger than 0.5 kcal mol⁻¹ for all amino acid combinations (wild type and mutant). **d,e**, Thermodynamic coupling for N-terminal OB-domain of SO1732 (PDB ID: 2KCM) Y8–E38 (**d**) and α-spectrin SH3 domain (PDB ID: 1QKX) Y10–Y52 (Y15–Y57 in the PDB structure) (**e**). Left, domain structures showing the two mutated positions. Second from left, stabilities ($\Delta G$) of all pairs at the two mutated positions. Second from right, agreement between stabilities from the additive model ($x$ axis) and the observed stabilities ($y$ axis, wild-type pair shown in red). Right, structural models of mutant pairs with strong couplings. Couplings

show the observed stability minus the expected stability from the additive model; uncertainties show the s.d. from computing the couplings using 1,000 bootstrapped samples of the 400 double mutants. **f**, Thermodynamic coupling without a visible sidechain interaction in the MYO3 SH3 domain D10–K24 (D1131–K1145 in PDB ID 2BTT). Sub-panels as in **d**, with the magnified view of the wild-type structure of Y9, D10 and K24 shown on the right. **g**, Thermodynamic coupling mediated by a third amino acid in the J domain of HSJ1a: Y3–R60–D64 (Y5–R62–D66 in PDB ID 2LGW). Left, the AlphaFold-modelled structure of the HSJ1a J domain with three interacting amino acids. Scatter plots show the stabilities of double mutants in the additive model ($x$ axis) and experimental data ($y$ axis) in the wild-type background (blue) and with the third residue replaced by Ala (orange). Right, the thermodynamic coupling for each pair of wild-type amino acids in the wild-type (blue) and the Ala-substituted (orange) backgrounds (error bars represent the s.d. from bootstrap resampling ($n = 1,000$) as in **d**). Substituting any of the three amino acids for Ala eliminates the coupling between the other two.

and the background in which the third amino acid was replaced by alanine (400 mutants × 3 pairs × 2 backgrounds ≈ 2,400 mutants in total for each triplet). As before, we modelled each set of 400 mutants (that is, one residue pair in one background) using 40 single-amino-acid parameters (we did not globally model all 2,400 mutants together). One notable triplet is found in the J domain of HSJ1a, where R60 and D64 both interact with the hydroxyl group on Y3 (Fig. 4g, left). We observe strong couplings (more than 1.5 kcal mol$^{-1}$) between each pair of 2 out of the 3 amino acids. However, when any of the 3 amino acids is mutated to alanine, the coupling between the remaining two amino acids becomes much weaker (less than 0.5 kcal mol$^{-1}$; Fig. 4g, middle and right and Supplementary Fig. 4d). These results reveal a strong third-order coupling: the interaction between two amino acids is mediated by a third amino acid.

This strong three-way coupling is noteworthy because the interactions do not appear in the deposited NMR ensemble (PDB ID: 2LGW; Extended Data Fig. 8a,b). The NMR ensemble for 2LGW positions Y5 (Y3 in our numbering) away from the helix containing R62 and D66, making the interaction network impossible. However, the AlphaFold-predicted structure shown in Fig. 4g (the highest confidence model out of five predictions) does include these interactions, which are also seen in other J-domain crystal structures from *Caenorhabditis elegans* (PDB ID: 2OCH) and *Plasmodium falciparum* (PDB ID: 6RZY). The strong couplings that we identify support the AlphaFold model and suggest the deposited ensemble is missing conserved interactions that form in HSJ1a, perhaps owing to the specific experimental conditions used. This example illustrates how large-scale folding stability measurements can reveal the thermodynamic effects of a critical interaction even when that interaction is not present in the deposited NMR structure. Notably, AlphaFold itself does not always predict this network either, depending on the specific linkers used (Extended Data Fig. 8d,e).

The scale of cDNA display proteolysis makes it straightforward to characterize unique cases such as these, which can serve as stringent tests for models of folding stability. Strong third-order couplings like this example also present a special challenge for computational models that calculate stabilities by summing interaction energies between pairs of residues using a single reference structure. Deep learning models that implicitly represent conformational landscapes[31] may be more promising, but training these models using large-scale thermodynamic measurements will be essential to achieve their potential.

## Influence of stability on evolution

Next, we examined how selection for stability influences protein sequence evolution in concert with other evolutionary mechanisms. It is well known that proteins contain specific functional residues that are commonly deleterious to stability[34,35]. However, the challenge of measuring stability has made it difficult to experimentally distinguish selection for stability from other selective pressures on a global level[36–38]. To examine the strength of selection for stability, we created a simple classification model to predict the wild-type amino acid at any site in a natural protein based on the folding stabilities of all substitution variants at that site (excluding Cys) (Fig. 5a). The model contains two parts: (1) a shared weight function that converts absolute stabilities of protein variants into relative probabilities of those amino acids, and (2) amino-acid specific offsets that shift amino acid probabilities by a constant amount at all sites. We fit the parameters of the shared weight function (a flexible monotonically increasing function) and the offsets together using stability data for wild-type sequences and substitution variants at 5,214 sites in 90 non-redundant natural proteins (99,156 ΔG measurements in all; Fig. 5a). Our simple model fits the data well by three criteria: (1) it correctly produces the overall frequencies of the 19 (non-Cys) amino acids (Fig. 5b), (2) the predicted amino acid probabilities are correctly calibrated across the full range of probability (Supplementary Fig. 5), and (3) the model performs similarly well on the training set and on a held-out testing set consisting of 758 sites in 11 domains with no similarity to the training set (Fig. 5e).

The model parameters reveal the strength of selection for stability across this diverse set of domains from many organisms. Within the main range of our data (folding stabilities from 1.5 to 4 kcal mol$^{-1}$), amino acid probabilities increase approximately linearly with increased stability, with a 1 kcal mol$^{-1}$ stability difference between protein variants indicating an approximately 9.2-fold difference in sequence likelihood (Fig. 5c). The slope is steeper in the low-stability region (ΔG < 1 kcal mol$^{-1}$), indicating stronger selection for stability. However, our 90-protein training set includes only 2 wild-type sequences with ΔG < 1 kcal mol$^{-1}$, and this may bias this result. The global offsets to each amino acid's probability (Fig. 5d) are different from the empirical amino acid frequencies (Fig. 5b) and indicate the probabilities of each amino acid under conditions in which all sequence variants are equally stable. The offsets span a 32-fold range: the most likely amino acid (Glu) is 32-fold more likely to occur ($2^{1.5}/2^{-3.5}$) than the least likely amino acid (Trp) when sequence variants with these amino acids at the same site are equally stable (Fig. 5d). This probability difference corresponds to a stability difference of ~1.6 kcal mol$^{-1}$ (Fig. 5c); that is, Trp and Glu would be equally likely at a site if the Trp variant were 1.6 kcal mol$^{-1}$ more stable than the Glu variant. Assuming equal stabilities, the most likely amino acids are the charged amino acids Glu, Asp, and Lys, suggesting selection for solubility, whereas the least likely amino acids are the nonpolar aromatic amino acids Trp, Phe and Tyr, along with Met. These offsets provide a quantitative 'favourability' metric incorporating all non-stability evolutionary influences on amino acid composition, including selection for amino acid synthesis cost[39], codon usage[40], avoiding oxidation-prone amino acids, net charge and function. These offsets also highlight that biophysical models and protein design methods trained to reproduce native protein sequences will not consistently optimize folding stability; Fig. 5d quantifies the amount by which specific amino acids are over- or underrepresented in small domains compared to their effects on stability. Notably, these offsets are similar to findings from an independent analysis of global discrepancies between variant effect data and sequence-likelihood modelling[41].

## Properties of functional residues

Selection for function also causes protein sequences to diverge from the variants with the highest stability sequence. Previous studies[36,37] used this principle to identify functional sites based on the difference between evolutionary conservation and predicted effects on stability. We expanded this strategy to use experimental stability measurements and examined the properties of functional sites on a large scale. We identified functional sites in 104 diverse protein domains by comparing each site's average ΔΔG of substitutions with its normalized GEMME[42] score, a measure of sensitivity to mutations inferred from multiple sequence alignments (Extended Data Fig. 9a and Methods). High sensitivity generally indicates high evolutionary conservation. Sites where wild-type amino acids are critical for stability (more negative average ΔΔG, rightward) tend to be more sensitive to mutation (upward) and vice versa. We defined all sites in the upper left region (where the site is sensitive to mutations yet unimportant for stability, 9% in total) to be 'functional' sites. This classification correctly identifies key binding residues in the chromodomain of HP1 and the SH3 domain of BBC1 (Extended Data Fig. 9b,c, full data in Supplementary Fig. 6). Across all 104 domains, Gly, Asp and the bulky amino acids (Trp, Arg and Tyr) were frequently classified as functional (Extended Data Fig. 9d), and the fraction of functional sites ranged from 0 to approximately 25% (Extended Data Fig. 9e). The domains with the highest fraction of functional sites were the nucleic acid binding domains Sso7d (PDB ID: 1JIC) and ribosomal protein S19 (PDB ID: 1QKH; Extended Data Fig. 9f).

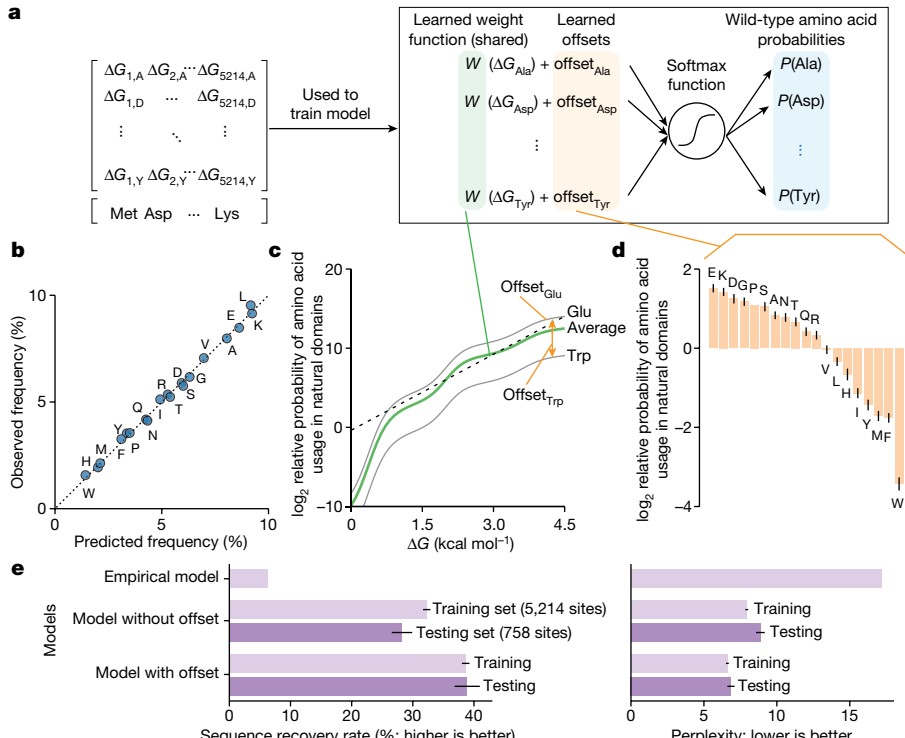

**Fig. 5 | Amino acid frequencies in natural proteins systematically deviate from maximizing stability. a**, Classifier model for predicting wild-type amino acids from the folding stabilities ($\Delta G$) of each protein variant. A shared weighting function converts the stabilities of protein variants with each amino acid into probabilities of those amino acids occurring (green). The probability for each amino acid is further modified by a constant amino acid-specific offset (orange). The model was parameterized using 99,156 $\Delta G$ measurements (5,214 sites in 90 non-redundant natural domains). **b**, Predicted and observed amino acid frequencies according to the classifier model after fitting. Amino acid frequencies were calculated for 5,214 sites. **c**, The weighting function from the classifier model after fitting (green). Grey lines show the weighting function after amino acid-specific offsets for Glu and Trp. In the region between 1.5 and 4 kcal mol⁻¹, the function has an approximately constant slope whereby a

1 kcal mol⁻¹ increase in stability leads to a 9.2-fold increase in amino acid probability (indicated by a dotted line). **d**, Relative offsets for 19 amino acids from the classifier model after fitting. Error bars show the s.d. of the model posterior ($n = 25$). At sites where $\Delta G_{Val} = \Delta G_{Trp}$, valine is around 11 times ($= 2^0/2^{-3.5}$) more likely to be the wild-type amino acid. **e**, The sequence recovery rate (left) and perplexity (right) for predicting wild-type amino acids using several models: a null model that ignores stability and always predicts amino acids at their observed frequencies, our classifier model without amino acid-specific offsets, and our full classifier model. Similar performance of the classifier model on a training set of 5,214 positions (light purple) and a testing set of 758 positions (dark purple) indicates that the model is not overfitted. Error bars show s.d. from bootstrap resampling ($n = 1,000$).

Our original criteria classify nearly all buried sites as non-functional; the mutational sensitivity at these sites is attributed to the destabilizing effects of hydrophilic substitutions. To identify buried functional sites, we modified our criteria by considering only substitutions to nonpolar amino acids (Extended Data Fig. 9g). With this approach, most functional sites are still on the surface, but some are found in the core (Extended Data Fig. 9h). In the DUF1471 domain of *Salmonella typhimurium* yahO, GEMME's evolutionary model indicates that the buried A64 is sensitive to nonpolar substitutions, but our data show that substitutions to Tyr or Phe increase folding stability (Extended Data Fig. 9i). This suggests that A64 is important for function, perhaps by maintaining the overall protein shape. Similarly, in the N-terminal domain of human FK506-binding protein 3, GEMME's model indicates that the buried L55 is sensitive to nonpolar substitutions, but our data show that substitutions to Ile, Val or Phe have no effect on stability (Extended Data Fig. 9j). Again, this suggests that L55 is important for function and that substitutions at L55 may allosterically modulate the domain's DNA binding activity. This is consistent with NMR experiments that show a chemical shift perturbation at L55 in response to DNA binding, even though the residue is buried beneath the surface[43] (Extended Data Fig. 9j). The other chemical shift perturbations are mainly found in the functional residues on the surface. These results highlight unusual cases where buried sites are conserved for function instead of stability.

## Large-scale data to guide design

The unique scale of cDNA display proteolysis creates new opportunities for improving protein design. Here, we examined three applications: (1) characterizing the stability determinants of highly polar designs, (2) identifying stabilizing mutations, and (3) benchmarking the protein design tool PROSS[44]. The hydrophobic effect is considered the dominant force in protein folding[2] and previous studies of designed miniproteins have emphasized the importance of nonpolar burial[18,30]. We hypothesized that mutational scanning of high-stability, highly polar designs could reveal alternative routes to high stability while minimizing hydrophobicity-induced aggregation. Although the mutational scanning patterns for highly polar designs were not obviously different from other designs, we identified several designs that possessed exceptionally strong polar interactions (large dots in Extended Data Fig. 10a). In Extended Data Fig. 10b, we highlight stabilizing polar networks and a cation–π interaction in these unusual designs (see Supplementary Fig. 7 for full mutational scanning results). The average $\Delta\Delta G$ for substitutions at these polar sites ranged from −0.20 to −1.33 kcal mol⁻¹, corresponding to the top 63 to 1.5 percentile for all 3,694 polar sites in 145 designs. Our massive dataset made it possible to identify these rare highly stabilizing interactions. Notably, the second hydrogen bond network in EHEE_rd2_0152 is also found in two other more hydrophobic designs. However, the network is less sensitive to

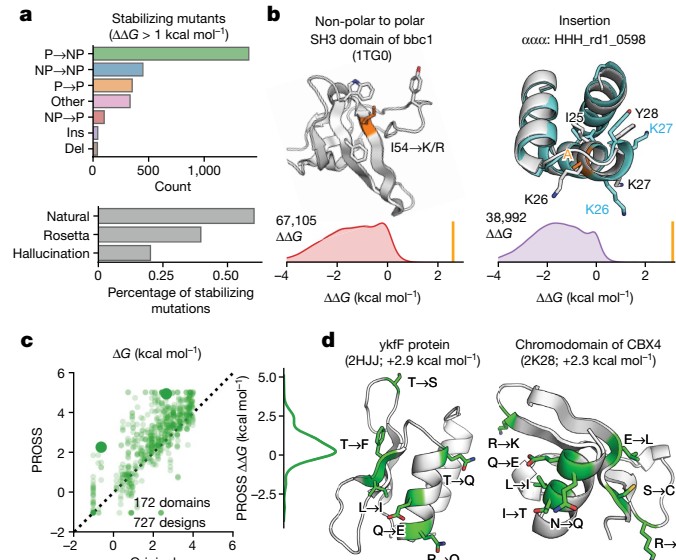

**Fig. 6 | Application of large-scale data to protein design. a**, Stabilizing mutations ($\Delta\Delta G > 1$ kcal mol$^{-1}$) found in natural domains, Rosetta designs and hallucination designs, broken down by mutation type. NP, non-polar; P, polar; ins, insertion; del, deletion. Data are from 448,788 mutants in 412 domains. **b**, Two examples of stabilizing mutations found by our assay, along with the distribution of $\Delta\Delta G$ values for these mutation types. The highlighted mutations are indicated by vertical bars. Full mutational scanning results are shown in Supplementary Fig. 8. The structure of HHH_rd1_0598 is a design model reported previously[18], not an experimental structure. **c**, Stabilities of test domains before (*x* axis) and after (*y* axis) redesign by PROSS[44]. The dashed black line represents *y* = *x*. Large dots indicate examples shown in **d**. Right, the distribution of $\Delta G$ change following PROSS redesign. Forty per cent of domains are stabilized by more than 1 kcal mol$^{-1}$ by the tool. **d**, AlphaFold models of domains redesigned by PROSS, with mutated amino acids in green. PDB ID and the change in stability are shown in brackets.

substitution in those designs, highlighting how the overall protein environment mediates the effects of substitutions even on the protein surface (Extended Data Fig. 10c–e).

We next examined how our approach could identify stabilizing mutations. Predicting and designing stabilizing mutations is a major goal of protein modelling, but prediction accuracy remains low[14]. In part, this is because stabilizing mutants are rare in current databases[11] (outside of reverting a destabilizing mutant), limiting the data available for improving modelling. Our large-scale approach revealed 2,600 mutations that increase folding stability by at least 1 kcal mol$^{-1}$. The fraction of stabilizing mutations was approximately 0.2% to 0.6% for different protein types (Fig. 6a). Stabilizing mutations were enriched at functional sites (23% of the stabilizing mutations from 8% of sites classified as functional). Notably, our set includes 112 examples of stabilizing insertions and deletions. Figure 6b shows two examples of different classes of stabilizing mutations from our dataset (Supplementary Fig. 8).

Finally, we applied our method to evaluate PROSS[44], an automated method for increasing folding stability using structural modelling and evolutionary data. We tested 727 PROSS designs for 172 protein domains with wild-type $\Delta G < 4$ kcal mol$^{-1}$. Unlike previous studies[45], our mutational scanning data enabled us to examine the isolated effect of every individual substitution in each design. The average increase in stability from PROSS was $0.6 \pm 1.0$ kcal mol$^{-1}$ (mean ± s.d.) (Fig. 6c). As intended, PROSS avoided mutations at functional positions: only 2% of PROSS-designed mutations were at functional sites, compared with 9% of sites classified as functional (defined in Extended Data Fig. 9a). Two example designs are shown in Fig. 6d. Larger numbers of mutations

typically led to larger increases in stability (Extended Data Fig. 11a), as theorized previously[14]. Our mutational scanning data showed that the average effect of a single designed mutation was $0.2 \pm 0.5$ kcal mol$^{-1}$ (Extended Data Fig. 11b). On average, the stabilization from PROSS was comparable in size to the best single designed mutation, and smaller than the two best mutations added together (Extended Data Fig. 11c). Evaluating individual designed mutations by direct comparisons to mutational scanning data provides a novel approach for systematically improving design methods.

## Discussion

The cDNA display proteolysis method massively expands the scale of folding stability experiments. Nonetheless, the method currently has notable limitations. First, our assay is limited to proteins that express and fold in the cell-free environment and are compatible with cDNA display. Owing to the very low concentration of each library member, proteins that are stabilized in complexes are probably unsuitable for this assay. Second, because we digest proteins under native conditions, our inferred thermodynamic stabilities are only accurate when (1) folding is fully cooperative (no segments get cleaved without global unfolding[46]), (2) folding is at equilibrium during the assay (no kinetic stability or spurious stability owing to aggregation), (3) $K_{50,U}$ is accurately inferred (Fig. 1c), (4) cleavage leads to dissociation of the cDNA (minimal disulfide or other crosslinking that could retain the C terminus after proteolysis), and (5) cleavage rates fall within the measurable range of the assay, which currently limits the dynamic range to around 5 kcal mol$^{-1}$ (Fig. 1c). Many domains—particularly larger protein structures—do not satisfy these conditions. Our mutational scanning data often suggested cases of non-cooperativity or aggregation, but these potential artefacts can be invisible when assaying individual sequences without mutational scanning. Furthermore, cleavage from folded states may be undetectable even with mutational scanning data if both proteases are equally affected and the cleavage is not overly sensitive to any individual mutation. Combining cDNA display proteolysis with chemical denaturation (pulse proteolysis[21]) may overcome some of these obstacles and enable mega-scale analysis of less cooperative and/or higher stability proteins. Advances in DNA synthesis[47,48] will also make it possible to expand cDNA display proteolysis beyond the largest domains studied here (72 amino acids). Finally, multiplexed measurements and automated data processing have the potential to introduce inaccuracies, although we worked to exclude unreliable data. For notable individual results, examining the raw data can be helpful, and we included all data and code to regenerate all fits.

Despite these limitations, the unique scale of cDNA display proteolysis opens completely new possibilities for studying protein stability. By comprehensively measuring single mutants across nearly all small structures in the Protein Data Bank, we quantified several global trends: trends in amino acid fitness at different sites, trends in the effects of single and double mutants, and trends in how stability influences sequence evolution. Alongside these trends, our analysis uncovered hundreds of exceptional cases that would be challenging to identify by smaller-scale methods. These include mutations with extreme effects, sites with unusual stability landscapes, and pair interactions with unusually strong thermodynamic couplings. The thermodynamic couplings that we identified in the J domain of human HSJ1a (Fig. 4g)—which was not present in the deposited NMR structure—highlight how large-scale stability assays can complement other methods for revealing structural details in solution. Beyond studying stability, cDNA display proteolysis will have other applications, including assaying designed proteins on a massive scale to systematically improve design methods[18,30,31,49] and to dissect the relationships between folding stability and function[50].

Achieving an accurate, quantitative understanding of protein stability and its sequence dependence has long been a central goal in

biophysics. We envision millions of cDNA display proteolysis measurements forming the foundation for a new generation of deep learning models that predict absolute folding stabilities and effects of mutations. Breakthroughs in structure prediction powered by deep learning have demonstrated the power of these models in protein science, but collecting sufficient thermodynamic data has always been a major obstacle. Owing to the scale and efficiency of cDNA display proteolysis, the main limit to measuring stability for millions of small domains is the cost of DNA synthesis and sequencing—both of which are rapidly decreasing. The size and diversity of the protein sequence space creates enormous challenges for biology and protein design. cDNA display proteolysis offers a powerful approach for large-scale mapping of folding stability across this space.

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

# Methods

## Selection of natural proteins for mutational scanning

We first collected all monomeric proteins in the PDB in the 30–100 amino acid length range in June 2021. We next excluded structures that had only a single helix, contained other molecules (for example, proteins, nucleic acids or metals), were annotated to have DNAse, RNAse or protease inhibition activity, or included more than four cystines. We then removed redundant sequences (amino acid sequence distance <2). We then predicted the structures of these PDB sequences using AlphaFold (even though the PDB structures were known), and used the AlphaFold models to trim amino acids from the N- and C termini that had a low number of contacts with any other residues. Finally, we selected domains with up to 72 amino acids after excluding N- or C-terminal flexible loops.

## EEHH design method

EEHH protein design was performed in three steps: (1) backbone construction, (2) sequence design, (3) selection of designs for deep mutational analysis. Backbone construction (the de novo creation of a compact, three-dimensional backbone with a pre-specified secondary structure) was performed using a blueprint-based approach described previously[51,52]. All blueprints are included as Blueprints_for_EEHH.zip in Source data.

## Hallucination design method

We used a TrRosetta hallucination protocol described previously in the previous reports[31,32] and available at https://github.com/gjoni/trDesign/tree/master/02-GD to unconditionally generate protein backbones and sequences with lengths ranging from 46 to 69 amino acids by maximizing the Kullback–Leibler divergence between the predicted and background distance/angle distributions. Predicted distograms and anglegrams were used to obtain 3D structures of these models as described in the TrRosetta paper[53]. We selected the best designs according to the predicted distogram and 3D structure match.

## DNA oligonucleotide library construction

All sequences were reverse-translated and codon-optimized using DNAworks2.0[54]. Sequences were optimized using *E. coli* codon frequencies because we used an in vitro translation kit derived from *E. coli*. Oligonucleotide libraries encoding amino acid sequences of Library 1 were purchased from Agilent Technologies.

**Library 1.** We selected ~250 designed proteins and ~50 natural proteins that are shorter than 45 amino acids. Then, we created amino acid sequences for deep mutational scanning followed by padding by Gly, Ala and Ser amino acids so that all sequences have 44 amino acids. The total number of sequences is ~244,000 sequences Purchased from Agilent Technologies, length 230 nt.

**Library 2.** We selected ~350 natural proteins that have PDB structures that are in a monomer state and have 72 or less amino acids after removing N and C-terminal linkers. Then, we created amino acid sequences for deep mutational scanning followed by padding by Gly, Ala and Ser amino acids so that all sequences have 72 amino acids. The total number of sequences is ~650,000 sequences. This library also includes scramble sequences to construct unfolded state model. Purchased from Twist Bioscience, length 250 nt.

**Library 3.** We selected ~150 designed proteins and created amino acid sequences for deep mutational scanning of the proteins. We also included comprehensive deletion and Gly or Ala insertion of all wild-type proteins included in Library 1 and Library 2. Additionally, amino acid sequences for comprehensive double mutant analysis on polar amino acid

pairs were also included. The total number of sequences is ~840,000 sequences. Purchased from Twist Bioscience, length 250 nt.

**Library 4.** Amino acid sequences for exhaustive double mutant analysis on amino acid pairs located in close proximity were included. We also include overlapped sequences to calibrate effective protease concentration and to check consistency between libraries. The total number of sequences is ~900,000 sequences. Purchased from Twist Bioscience, length 300 nt.

## DNA and mRNA preparation for cDNA display proteolysis method

Oligonucleotide libraries were amplified by PCR using KOD PCR Master Mix (Toyobo) to add T7 promoter, PA tag to an N-terminal, and His tag to an C-terminal of the proteins. The number of cycles was chosen based on a test qPCR run to avoid overamplification using SsoAdvanced Universal SYBR Green Supermix (Bio-Rad). The PCR product was gel extracted to isolate the expected length product. Then we used T7-Scribe Standard RNA IVT Kit (Cellscript) to synthesize mRNA using the DNA fragment as a template.

## Preparation of protein–cDNA complex

We followed the protocol essentially as described[24,55], with some modifications, described below.

**Photo-crosslinking between mRNA and the puromycin linker.** We prepared the photocrosslinking reaction solution (usually at 40 µl scale) using 100 mM NaCl, 20 mM Tris-HCl (pH 7.5), 1 µM cnvK linker (EME), 1 µM mRNA. The solution was incubated at 95 °C for 5 min, then slowly cooled down to 45 °C (0.1 °C s⁻¹) using a thermal cycler. Then the solution including the duplex was irradiated with UV light at 365 nm using a 6 W Handheld lamp (Thermofisher) for 15 min. At 40 µl scale (40 pmol cnvK linker and 40 pmol mRNA total), this produces crosslinked mRNA sufficient for 48 proteolysis reactions.

**In vitro translation and reverse transcription.** We used the PUREfrex 2.0 (GeneFrontier) translation system according to the manufacturer protocol. We typically used a 160 µl total reaction including 40 µl of the mRNA-cnvK linker duplex product from Step 1 and RiboLock RNase Inhibitor (Thermofisher). We incubated the reaction at 37 °C for 2 h. After the incubation, 500 mM EDTA (16 µl for a 160 µl reaction) was added to the sample to dissociate ribosomes. Then, an equal amount (160 µl for a 160 µl reaction) of 2× binding/washing buffer (20 mM Tris pH 7.5, 2 mM EDTA, 2 M NaCl, 0.2% Tween) was added. The solution was added to Dynabeads MyOne Streptavidin C1 (Thermofisher, 200 µl for 40 pmol mRNA) to pull down the protein-mRNA complex and incubated at room temperature for 20 min. Before use, streptavidin beads were pre-washed with (1) 100 mM NaOH, 50 mM NaCl, then (2) 100 mM NaCl to remove any RNase activity. After streptavidin pull-down, the beads were washed by 1× binding/washing buffer once and rinsed twice by TBS (10 mM Tris-HCl pH7.5, 100 mM NaCl), and we added reverse transcription solution (PrimeScript RT Reagent Kit; Takara) onto the beads with protein mRNA complex, and incubated the beads at 37 °C for 30 min.

**Purification of protein–cDNA complex.** After the reverse transcription, the protein–cDNA complex was eluted with His-binding buffer (30 mM Tris pH7.4, 0.5 NaCl, 0.05% Tween) with RNase T1 (Thermofisher) usually in 400 µl scale. The eluent was added to His Mag Sepharose Ni (Cytiva) (800 µl for 40 pmol starting mRNA) and incubated at room temperature for 30 min. Then the complex was eluted by His-binding buffer with 400 mM imidazole (usually 400 µl) and the eluent was buffer-exchanged to PBS by Zeba Spin Desalting Column (Thermofisher). Then the complex was snap-frozen with liquid nitrogen and stored at −80 °C until the following protease assay. When starting from

40 pmol cnvK linker and 40 pmol mRNA for step 2, we would typically finish this step with 400 µl of protein–cDNA complex divided into 4 frozen tubes (100 µl each) for four sets of 12 protease experiments (48 conditions total).

## Protease assay on protein–cDNA complex

Proteolysis reactions were performed in two 'replicates' of 12 conditions each (11 protease concentrations in a threefold dilution series and one condition with no protease). Replicate 1 used a maximum protease concentration of 25 µM and replicate 2 used 43.3 µM ($25 \times 3^{0.5}$ µM). For one replicate (12 reactions), we started from ~25 µl complex, diluted this in PBS up to 240 µl, then added 20 µl to each of the 12 Protein LoBind tubes used for that replicate. Each reaction contained protein–cDNA complex equivalent to 0.83 pmol starting cnvK linker and 0.83 pmol starting mRNA. To start each reaction, we added 40 µl of protease solution to each tube. After 5 min protease digestion at room temperature, we added 200 µl chilled 2% BSA in PBS to quench the reaction, then the solution was added to 40 µl Dynabeads Protein G (Thermofisher) preincubated with anti-PA tag (Wako; Clone number: NZ-1; 1 µg antibody per 30 µl beads), and incubated at 4 °C for 1 h. Then the beads were washed by washing buffer (PBS including 800 mM NaCl and 1% Triton) three times and rinsed by PBS three times, then the complex was eluted with 50 µl PBS including 250 µg ml$^{-1}$ PA peptide (Wako) and 200 µg ml$^{-1}$ BSA (Thermofisher). Trypsin experiments used Trypsin-EDTA (0.25%) with phenol red (Thermo Fisher Scientific) for consistency with ref. 18 and chymotrypsin experiments used α-chymotrypsin from bovine pancreas (Sigma).

## qPCR analysis of cDNA display proteolysis results on individual proteins

The cDNA amount for each specific sequence in the eluents was quantified by qPCR using SsoAdvanced Universal SYBR Green Supermix and specific primers for each sequence. The qPCR was performed using CFX96 Touch Real-Time PCR Detection System (Bio-Rad), and the qPCR cycles were determined by the CFX Maestro Software (Bio-Rad). Extended Data Fig. 1.

## Next-generation sequencing sample preparation

For DNA library analysis, one-half volume (25 µl) of the eluted cDNA of the complex was amplified by PCR using SsoAdvanced Universal SYBR Green Supermix (BioRad) to add P5 and P7 NGS adapter sequences. The number of cycles was chosen based on a test qPCR run using the same PCR reagents to avoid overamplification. The DNA fragment length and concentration were confirmed by 4200 TapeStation System (Agilent), then the samples were analysed by NovaSeq 6000 System (Illumina).

## Processing of next-generation sequencing data

Each library in a sequencing run was identified via a unique 6- or 8-bp barcode. Following sequencing, reads were paired using the PEAR program[56] then the adapter sequences were moved by Cutadapt[57]. Reads were considered counts for a sequence if the read perfectly matched the ordered sequences at the nucleotide level.

## Overall strategy for inferring $K_{50}$ and $\Delta G$ from sequencing data

We used Bayesian inference to infer $K_{50}$ and $\Delta G$ values for all sequences in our library. This analysis uses two main models. The first model is called the '$K_{50}$ model' and infers each sequence's $K_{50}$ values based on the sequencing count data. The second model is called the 'unfolded state model' and predicts each sequence's unfolded state $K_{50}$ value ($K_{50,U}$) based on its sequence. Both models are implemented in Python 3.9 using the Numpyro package[58] version 0.80. In Supplementary Notes, we describe the structure of each model and the procedure for fitting each model. Our scripts to reproduce the complete fitting process are provided in the Source Data.

## Replicate analysis of $K_{50}$

Instead of sampling $K_{50}$ values using 24 samples per protease at one time as described in step 5 above, we sampled $K_{50}$ values using one experiment set (that is, 12 samples) and obtained $K_{50}$ for trypsin replicates 1 and 2, and chymotrypsin replicates 1 and 2. Note that we still used the calibrated protease concentrations to improve consistency between replicates. The replicates were conducted on different days using the same preparation of the protein–cDNA complex.

## Data for purified protein experiments

The data on purified proteins shown in Fig. 1g was taken from refs. 29,59–71.

## Data quality filtering and classification of datasets 1–3

Our data (Fig. 2) were filtered for quality in three stages. First, our Bayesian procedure produces confidence intervals for $K_{50}$ and $\Delta G$ estimates, producing a quality estimate for each individual measurement. Second, we evaluated the quality of each full mutational scan, classified these into categories, and removed the low quality categories from our main analysis (below). Third, we filtered our mutational scanning data to remove mutants that showed evidence of causing cleavage from the folded state or intermolecular disulfide cross-linking.

**Analysis of Bayesian confidence intervals.** Nearly all low-confidence $\Delta G$ estimates result from stabilities that are outside the main dynamic range of the assay (−1 to 5 kcal mol$^{-1}$). This is due to the very steep slope of $\Delta G$ with respect to $K_{50}$ in this range (see Fig. 1e). For all figures, we clip all $\Delta G$ estimates to the range of −1 to 5 kcal mol$^{-1}$ before further analysis. In the table of all data, the 'dG_ML' column categorizes sequences as '<−1' and '>5' if the 95% confidence interval is fully outside the range. Of the sequences with $\Delta G$ estimates between −1 and 5 kcal mol$^{-1}$, the median sequence had a 95% confidence interval width of 0.14 kcal mol$^{-1}$, and 99.9% of sequences had confidence intervals smaller than 0.96 kcal mol$^{-1}$. Although a very small fraction of $\Delta G$ estimates were low confidence (that is, had a wide confidence interval), we still included these sequences in all analyses. Note that these confidence intervals only reflect the model's uncertainty stemming from the finite deep sequencing counts; other uncertainties (such as uncertainty in $K_{50,U}$, $K_{50,F}$, protease concentrations, the validity of the kinetic model, and so on) are not reflected in these confidence intervals.

**Classification of mutational scans.** All mutational scanning data were classified into 12 groups (0 to 11) according to the protocol in Extended Data Fig. 8. Groups 0 and 1 contain the mutational scans that passed all quality filters. Domains in group 0 have wild-type $\Delta G$ values below 4.75 kcal mol$^{-1}$ so that stabilizing mutations can still fall within the cDNA proteolysis assay's dynamic range. Group 1 contains the remaining high-quality domains. Groups 2–11 contain mutational scans that failed one or more quality filters. All mutational scans are included in only one group, so a mutational scan classified as 'group 5' (for poor correlation between independent trypsin and chymotrypsin results) might also fail other filters (such as having a poor slope or intercept between trypsin and chymotrypsin results).

Below, we define each group, along with a short explanation of possible causes.

Group 0: Passing all quality filters.

Group 1: Passing all quality filters, but wild-type $\Delta G > 4.75$ kcal mol$^{-1}$, so stabilizing mutants may not be resolved compared to the wild type.

Group 2: Poor expression in the assay, based on low counts in next-generation sequencing.

Group 3: The wild-type protein is too unstable to see sequence–stability relationships. This may be due to a truly unstable wild-type sequence or due to rapid cleavage of some segment in the folded state.

Group 4: The wild-type stability (ΔG) is inconsistent. We often observed this for high stability proteins in our first library where the wild-type stability exceeded the dynamic range of the assay.

Group 5: Poor correlation between trypsin experiment and chymotrypsin experiments. This can suggest that one or both proteases are not probing global unfolding, leading to different mutational patterns between the proteases.

Group 6: Poor slope between trypsin experiment and chymotrypsin experiments. This can suggest that some cleavage is occurring from folded state(s) for one or both proteases. If cleavage can occur from the folded state for one protease, the modelled $K_{50,F}$ will be different from the true $K_{50,F}$, creating a slope between the inferred ΔG values and the true ΔG values (see Fig. 1e).

Group 7: Too many stabilizing mutants. In a typical well-folded domain, most mutations are neutral, so a very large fraction of stabilizing mutations suggests the wild-type ΔG may not have been measured accurately. Furthermore, when the large majority of hydrophobic substitutions at surface sites are stabilizing, this suggests the domain may be stabilized by non-specific intermolecular interactions. For these reasons, we removed domains showing these patterns.

Groups 8 and 9: Includes multiple cysteines with proper folding (G8) or misfolding (G9). Disulfide linkages have the potential to disrupt our assay by preventing the C-terminal cDNA from dissociating from the protein N terminus even after the protein is proteolysed. In general, we found that proteins with >1 Cys performed poorly in our assay and many of these proteins are found in groups 2–7. Owing to these results, we decided to remove the remaining proteins with >1 Cys (group 9). However, two proteins appeared to produce good results. Although we chose not to include these proteins in our main analysis, they have been separated into group 8 (high-quality data from proteins with >1 Cys).

Group 10: Poor intercept between trypsin experiment and chymotrypsin experiments. A poor intercept indicates that our trypsin and chymotrypsin experiments cannot agree on where ΔG = 0 is for the overall mutational scan. This depends on the unfolded model for each protease (the inference of $K_{50,U}$ for each protease). Because the two proteases did not agree on the ΔG values for these sequences, the ΔG values are likely less reliable than those in group 0 and group 1. However, ΔΔG values for this group are still consistent across both proteases.

Group 11: Probably cleavable in folded states. In many cases, excessive cleavage from the folded state or partially folded states will lead to low wild-type stability (G3), poor correlation between the proteases (G5), or a poor slope (G6). However, we saw some evidence of folded state cleavage even in mutational scans that passed these filtering criteria. Specifically, we observed cases where mutating out a wild-type cut site led to increased protease resistance (higher $K_{50}$) and apparently higher stability (ΔG) to one specific protease but not the other (for example, R16 in Extended Data Fig. 6a,b). This increase in apparent stability for just one protease suggests that either the site can be cleaved from folded state(s) for that protease, or removing the cut site is decreasing unfolded state susceptibility ($K_{50,U}$) in a way that is not properly accounted for by our model. Because these conditions lower the reliability of our ΔG estimates, we removed these mutational scans from analysis. The code to perform this filtering is provided (Data_quality_filtering_script.ipynb).

**Removal of individual mutants that may disrupt the assay.** In the previous stage, we filtered out entire domains; here, we filtered out data from individual mutants in domains that otherwise passed filtering (that is, were in group 0 or group 1). We focused on two specific types of mutations that could disrupt our assay. First, we filtered out data where introducing new cleavage sites into poorly structured regions of a protein resulted in apparent destabilization. Because these mutants are located in poorly structured sites, the apparent destabilization

may result from cleavage from folded or partially folded states. These mutants were identified based on (1) apparent destabilization from introducing the new cleavage site, and (2) a low variance in stability between the other amino acids, which indicates a poorly structured region of the protein where cleavage might occur in the folded state. Second, we filtered out data where introducing Cys mutants into poorly structured regions of a protein resulted in apparent stabilization. Again, because these mutants are located in poorly structured sites, the apparent stabilization may result from the formation of inter- or intramolecular disulfide linkages that prevent the dissociation of the C-terminal cDNA following protease cleavage. The code to perform this filtering is provided (Data_quality_filtering_script.ipynb).

All sequences in dataset 2 and dataset 3 are included in Tsuboyama2023_Dataset2_Dataset3_20230416.csv. All sequences in this file have an inferred ΔG estimate, but only sequences in dataset 3 have a tabulated ΔΔG estimate. Of course, one can calculate ΔΔG for the remaining sequences in dataset 2, but these ΔΔG values will be biased toward destabilizing mutations because stabilizing mutations would typically be indistinguishable from the wild-type stability. Note that datasets 2 and 3 include a very small number of sequences with low-quality data (wide confidence intervals) because these sequences come from mutational scans that are high quality overall. Although these tables include all $K_{50}$, ΔG and ΔΔG data (for dataset 3), low-quality data (including mutant data filtered in Stage 3) have been filtered out and replaced by a – symbol in the columns labelled '_ML' (for machine learning).

### Principal component analysis
We performed principal component analysis to determine the factors influencing stability of different amino acids (Fig. 3). To this end, we used 17,093 sites in the 365 domains that are classified as G0 in the above. All folding stability data were clipped between from −1 and 5 kcal mol$^{-1}$ because the folding stability outside the dynamic range is not reliable, and then the average of the stability for 20 amino acids for each site was subtracted from the data. Using the data, we performed principal components analysis using the scikit-learn library implemented in Python 3.

### Side chain contacts and burial analysis
Burial values and contact counts (Fig. 3b and Extended Data Fig. 9h) were computed based on AlphaFold models[1] of all sequences using the included script Burial_side_chain_contact_Fig3_Fig6.ipynb based on Bio. PDB[72] and BioPython[73]. The calculation is based on the Rosetta 'side-chain_neighbors' LayerDesign method previously reported[18]. In brief, to calculate the burial or contacts of residue X, we added up the number of residues in a cone projecting out 9 Å away from the Cβ atom on residue X in the direction of the residue X Cα-Cβ vector. 'Burial' (Fig. 6h) indicates the number of Cα atoms in the cone. Contact counts (Fig. 3d) each count different atoms inside the cone: 'side chain contact count' (Fig. 3d) counts all Cβ atoms; 'aromatic side chain contact count' counts all CE2 atoms of Phe, Tyr, and Trp; 'acidic side chain contact count' counts all Glu OE1 and Asp OD1 atoms; and 'Basic side chain contact count' counts all Lys NZ and Arg NE atoms.

### Secondary structure determination
Using the DSSP algorithm[74,75], we obtained secondary structure information based on AlphaFold models (Fig. 3b).

### Selection method of site pairs for double mutational analysis
Double mutants (Fig. 4) were selected for analysis in two ways. First, we manually selected polar interactions where either amino acid appeared important for stability in single mutational analysis. These pairs were mainly included in library 3. Second, we used the program confind[76,77] to identify interacting residues. All confind pairs with notable interactions such as polar interactions and cation−π interactions were selected, along with a randomly chosen subset of more common

interactions such as hydrophobic interactions. These pairs were included in library 4.

## Thermodynamic coupling analysis

The thermodynamic coupling model and the procedure for fitting the model (Fig. 4) are described in Supplementary Notes.

## Wild-type amino acid prediction model

The wild-type sequence prediction model (Fig. 5) and the procedure for fitting the model are described in Supplementary Notes.

## GEMME analysis

To calculate the normalized averaged GEMME score, which represents the sensitivity of a wild-type amino acid to substitutions inferred from evolutionary information ($\Delta\Delta E$ in the previous reports[36,37]), we ran GEMME[42] on each natural amino acid sequence using the default parameters. We computed a single score for each site by averaging the scores of the 19 amino acids (except Cys), and then standardized each domain individually (subtracted the domain's mean and divided by the domain's standard deviation) so that the site scores within a domain had a mean of zero and a standard deviation of one. Finally, we flip the sign of the score so that positive values imply high susceptibility to mutations (that is, very negative raw GEMME scores for non-wild-type amino acids). We define this standardized score for each site as the normalized GEMME score. To build the input multiple sequence alignments, we performed five iterations of the profile HMM homology search tool Jackhmmer[78,79] against the UniRef100 database of non-redundant proteins[80] using the EVcouplings framework[81]. We used the default bitscore threshold of 0.5 bit per residue.

## Structural modelling by AlphaFold

For most of the structural analysis, we used structural models predicted by AlphaFold[1]. We ran AlphaFold using default parameters and chose the model with the highest pLDDT score for each sequence. For designed sequences, we skipped a step for generating multiple sequence alignment.

## Statistics and reproducibility

We did not use statistical tests here. We did not perform multiple experiments under exactly the same conditions, but we used two different proteases and two different protease concentration sets to confirm reproducibility. In addition, we also confirmed that the same amino acid sequences show consistent $K_{50}$ values in different libraries (Extended Data Fig. 5).

## Reporting summary

Further information on research design is available in the Nature Portfolio Reporting Summary linked to this article.

## Data availability

All data are available in the main text, the extended data figures or tables, or available for download at https://doi.org/10.5281/zenodo.7992926.

## Code availability

The code for the analyses can be found at https://github.com/Rocklin-Lab/cdna-display-proteolysis-pipeline.

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

**Acknowledgements** The authors thank Epsilon Molecular Engineering (EME) for providing cnvK linker for cDNA display, Rush University and Genome Research Core at University of Illinois Chicago for performing next-generation sequencing, and D. Minh, T. Whitehead, K. Lindorff-Larsen, D. M. McCandlish, J. Maguire, J. Chodera, P. Hosseinzadeh, and the members of the Rocklin laboratory for discussions and comments on the manuscript. This work was supported by Northwestern University Startup Funding (G.J.R.), JSPS KAKENHI 19J30003 (K.T.), Human Frontier Science Program Long-Term Fellowship (K.T.) and JST PRESTO Grant JPMJPR21E9 (K.T.). This research was supported in part through the computational resources and staff contributions provided for the Quest high performance computing facility at

Northwestern University which is jointly supported by the Office of the Provost, the Office for Research, and Northwestern University Information Technology.

**Author contributions** K.T. designed and performed all experiments, and analysed the data with help from G.J.R. J.D. designed and analysed stabilities of hallucination-based proteins with help from S.O. J.C. designed ββαα proteins using Rosetta with help from G.J.R. E.L. computed GEMME scores with help from Y.M.B. and assisted with interpretation of GEMME. J.J.W. generated the PROSS designs. N.M.M. provided assistance with mathematical derivation and review of enzyme kinetic interpretation. G.J.R. and K.T. conceived the project. G.J.R. supervised the project and acquired funding. K.T. and G.J.R. wrote and revised the manuscript, with input from all authors.

**Competing interests** The authors declare no competing interests.

**Additional information**
**Correspondence and requests for materials** should be addressed to Gabriel J. Rocklin.

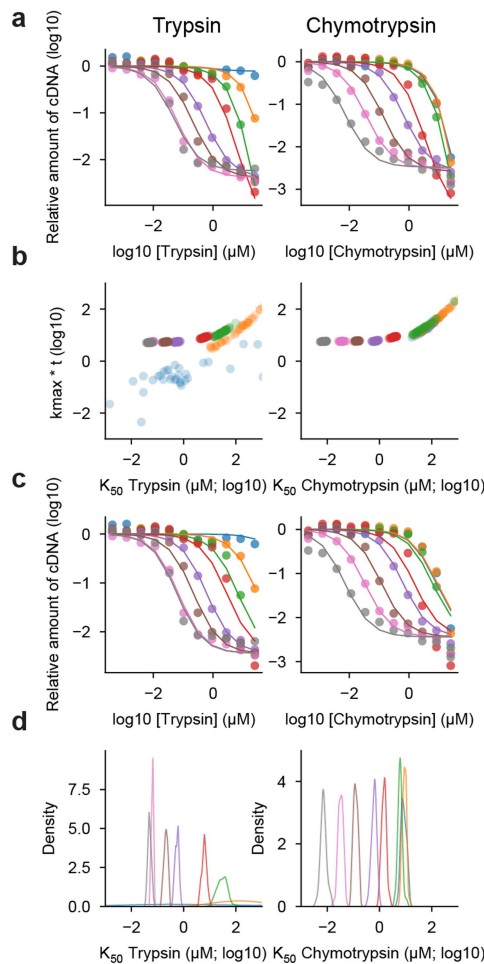

**Extended Data Fig. 1 | Single turnover model fitting on qPCR data. (a)** To test the single turnover model, we performed cDNA display proteolysis on a mixture of eight mini protein sequences with diverse folding stability and quantified the surviving amount of each cDNA using qPCR. We then each curve one at a time by Bayesian inference using the single turnover kinetics model in Fig. 1b. We sampled $k_{max}$*t and $K_{50}$ for each sequence. Dots represent the observed cDNA amount quantified by qPCR and lines show the two-parameter fits. **(b)** Posterior distributions of $k_{max}$*t and $K_{50}$ for eight proteins were shown. Whereas $K_{50}$ values vary between different proteins, $k_{max}$*t values (indicating saturation at high protease concentrations) were either constant or unconstrained by the data. **(c)** Based on the analysis (b), we fixed $k_{max}$*t at $10^{0.65}$ and re-sampled $K_{50}$ for each protein. Dots represent the observed cDNA amount quantified by qPCR (same as in (a)) lines show the one-parameter fits. **(d)** Posterior distributions of $K_{50}$. For trypsin, the $K_{50}$ values for the two most stable proteins (orange and blue) could not be defined because they were too stable and outside of the dynamic range of this proteolysis assay.

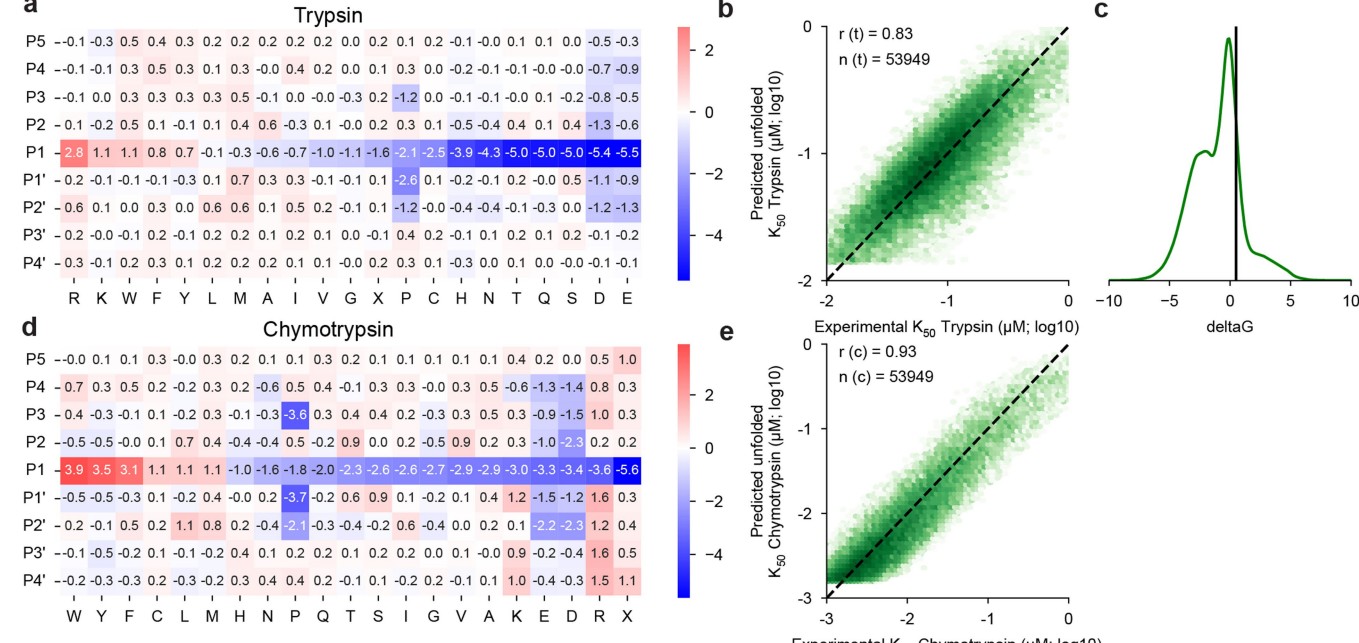

**Extended Data Fig. 2 | Unfolded state model parameters and goodness of fit.** (**a**) Fit parameters for the unfolded state model position-specific scoring matrix (PSSM) for trypsin. The mean of all coefficients (−0.4) was subtracted from the values in the figure to aid visualization. Positive values indicate faster proteolysis and lower predicted $K_{50,U}$ values. By using different prior distribution widths for different rows during fitting, we guided the strongest rate determinants into the center row of each matrix, which we label "P1" (the assay cannot actually identify the specific location of cutting). Overall, the heatmap resembles similar data as previously reported[18] and is consistent with known trypsin specificity determinants, including the preference for R/K at P1, the inhibitory effect of P, and the unfavorability of D and E[82]. (**b**) 2D-histogram showing the overall agreement between the trypsin model (predicted $K_{50,U}$, y-axis) and the data (experimental $K_{50}$, x-axis). Only scrambled sequences with inferred $\Delta G < 0.5$ kcal/mol (where we can assume $K_{50} \approx K_{50,U}$) are shown (53,949 out of 64,238 total sequences used in training). The Pearson r value is shown. (**c**) Overall distribution of inferred $\Delta G$ of all scramble sequences. The vertical line represents 0.5 kcal/mol, which is a threshold used in (b). (**d**, **e**) As above, for chymotrypsin. As in our previous report[18], the coefficients resemble established features of chymotrypsin specificity, including the preference for F/Y/W followed by M/L at P1, the inhibitory effect of P at P3, P1', and P2', and the general unfavorability of D and E[83–86]. The mean of all coefficients (−0.5) was subtracted from the values in the figure to aid visualization.

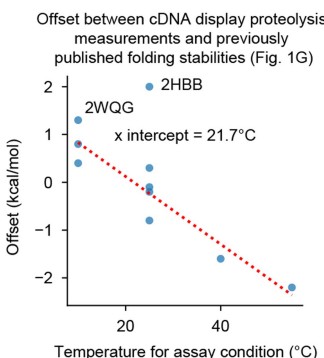

**Extended Data Fig. 3 | Relationship between offset in Fig. 1g and assay temperature.** Previous studies shown in Fig. 1g used diverse conditions including buffer, pH, ion strength, and temperature (see Supplementary Table 1)[29,59–71]. However, our measurements were all conducted in PBS at room temperature (approximately 22 °C). In general, the offsets observed in Fig. 1g are correlated to the temperatures used in the previous studies, suggesting that the assay temperature is the main cause of the offsets. The red line represents a best fit line. The x-intercept (21.7 °C) is close to our assay condition (approximately 22 °C). 2HBB (the N-terminal domain of Ribosomal Protein L9) and 2WQG (SAP domain from Tho1) were not included in the linear fit. 2HBB is an outlier; this may owe to its zinc-binding activity or to differences between the measured sequences (our construct lacks three C-terminal amino acids present in the previous study). 2WQG is close to the fit line but was removed because the previous literature used the L31W background as 'wild-type'; this mutation stabilizes the protein by 0.49 kcal/mol[60].

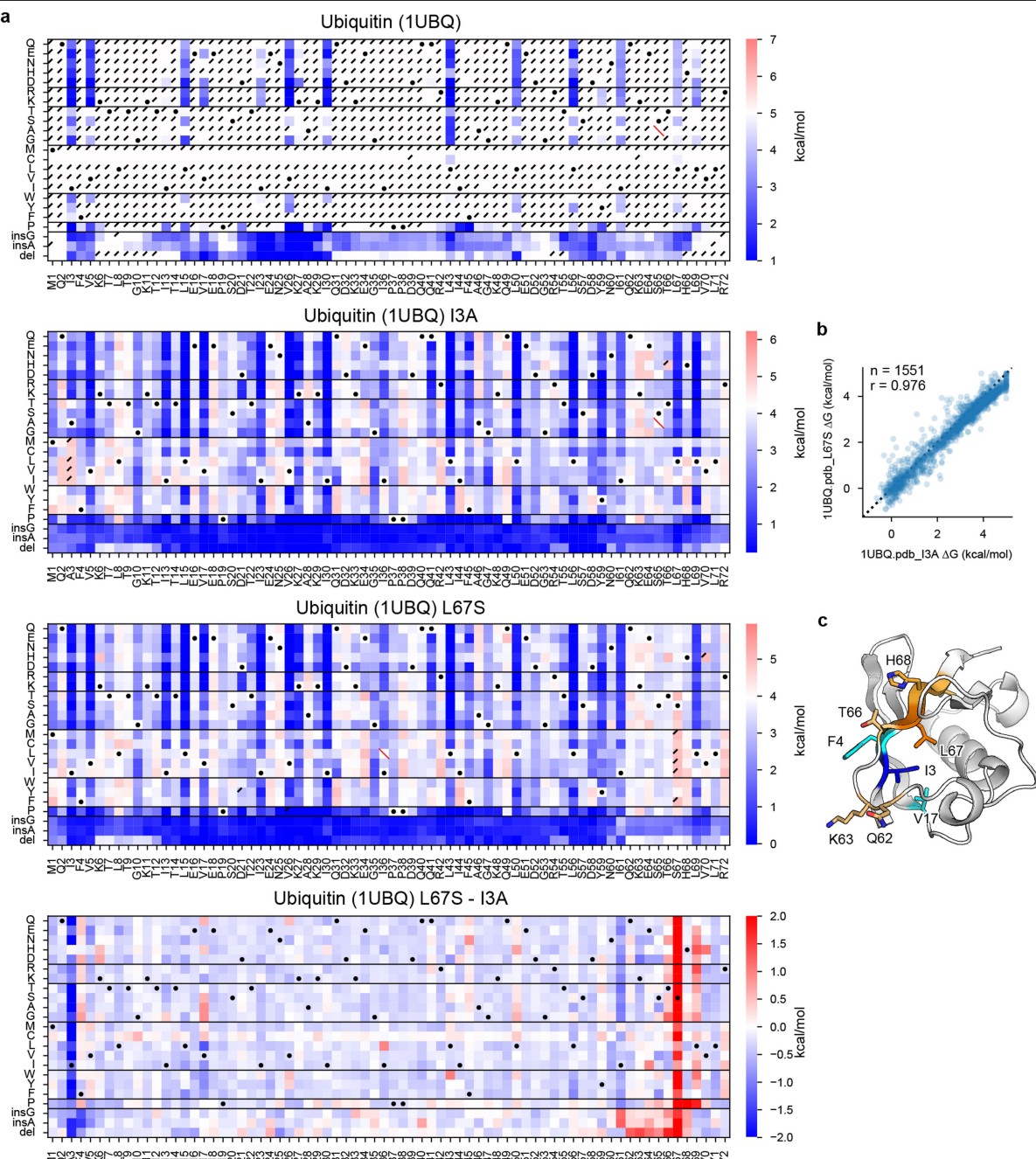

**Extended Data Fig. 4 | Heat maps for a stable domain (Ubiquitin; 1UBQ) and its destabilizing mutants.** (**a**) Mutational scanning results for human erythrocytic ubiquitin (1UBQ) and its destabilizing mutant backgrounds (I3A and L67S). Heat maps show the ΔG of wild-type ubiquitin (top), ubiquitin I3A (middle-top), ubiquitin L67S (middle-bottom), and the difference (ΔΔG) between two mutant backgrounds (bottom) for substitutions, deletions, and Gly and Ala insertions at each residue. In the three ΔG heat maps, white represents the folding stability of the wild-type and red/blue indicates stabilizing/destabilizing mutations. Black dots indicate the background

(wild-type or mutant) amino acid, red slashes indicate missing data, and black corner slashes indicate lower confidence ΔG estimates, (95% confidence interval > 0.5 kcal/mol), including ΔG estimates near the edges of the dynamic range. (**b**) Consistency between mutant stabilities measured in the I3A background (x-axis) and L67S (y-axis) background. The plot is annotated with the number of points and the Pearson r value. (**c**) Ubiquitin structure highlighting the mutant points (I3 and L67) and the residues with a different effect on stability between two mutational backgrounds.

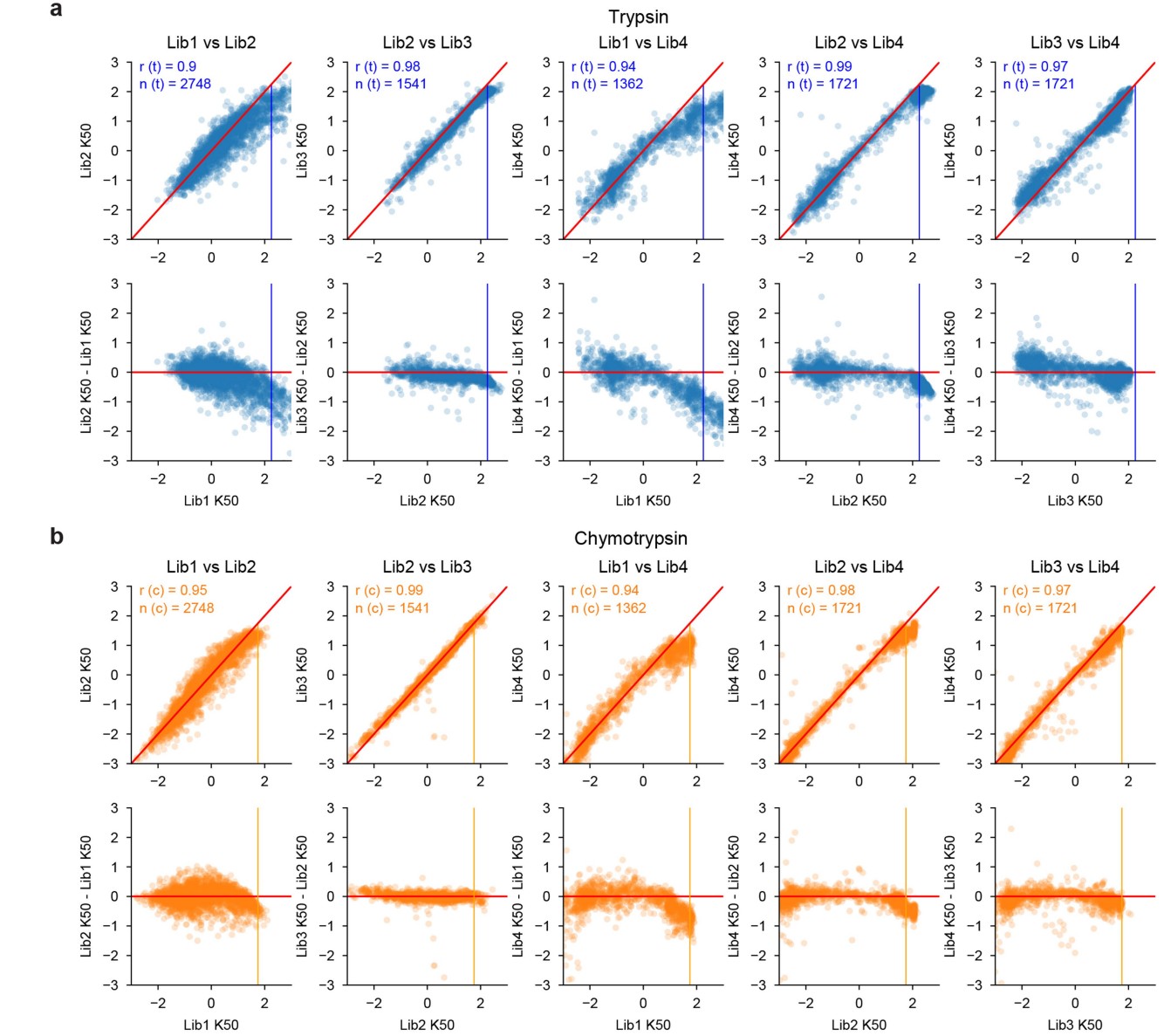

**Extended Data Fig. 5 | Consistency of $K_{50}$ measurements across libraries.**
(**a** and **b**) To examine the consistency between $K_{50}$ (µM) values measured in different libraries, we included identical sequences (potentially with different padding at the termini) in multiple libraries. For each pair of libraries with overlapping sequences, we show the $K_{50}$ values for those sequences in both libraries for trypsin (a) and chymotrypsin (b). The top row shows raw $K_{50}$ values for overlapping sequences in each library; the second row shows the difference in $K_{50}$ estimates plotted against the $K_{50}$ in one of the libraries. The red diagonal line shows Y=X in the top row and Y = 0 (i.e. identical $K_{50}$ estimates) in the bottom row. Blue/orange vertical lines show $K_{50,F}$; all $K_{50}$ values above $K_{50,F}$ are treated as equivalent. Each plot is annotated at the top-left with the total number of overlapping sequences and Pearson r-value between the libraries.

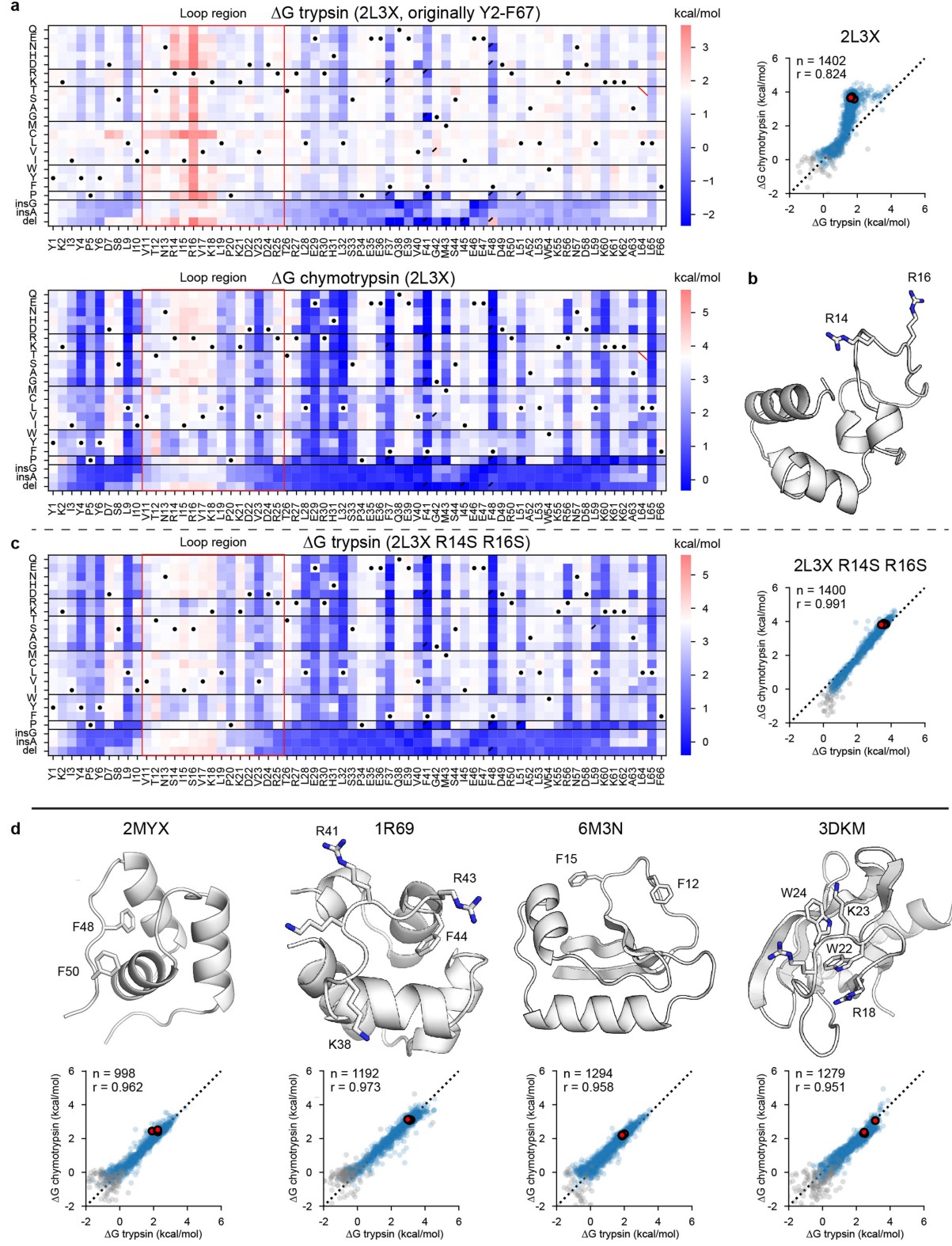

**Extended Data Fig. 6** | See next page for caption.

**Extended Data Fig. 6 | Domains with and without evidence of cleavage from the folded state.** (**a**) Mutational scanning results for 2L3X, which includes trypsin cleavage sites in the loop region. Left: Heat maps show the ΔG measurements from independent trypsin (top) and chymotrypsin challenges (bottom) for substitutions, deletions, and Gly and Ala insertions at each residue, with our one-indexed numbering at the bottom. Black dots indicate the wild-type amino acid, red slashes indicate missing data, and black corner slashes indicate lower confidence ΔG estimates, (95% confidence interval > 0.5 kcal/mol), including ΔG estimates near the edges of the dynamic range. The colored boxes highlight the flexible loop region. Right: Comparison of independent trypsin and chymotrypsin ΔG measurements. Multiple codon variants of the wild-type sequence are shown in red, reliable ΔG values in blue, and less reliable ΔG estimates (same as above) in gray. The black dashed line represents Y = X.

The dots show a reverse 'L' shape because trypsin can cleave the loop even from the folded state, lowering the apparent stability for the wild-type and all high-stability variants. (**b**) 2L3X structure highlighting arginines in the loop region (R14 and R16). (**c**) Same as (a) for 2L3X after removing the trypsin-cleavable sites (R14 and R16) from the loop. In this deep mutational scanning, we observed higher consistency between trypsin and chymotrypsin challenges because we removed sites that could be cleaved in the folded state. (**d**) Top: Four example domains with long protease-cleavable loops that do not show evidence of folded state cleavage. Bottom: Agreement between mutant ΔG values independently determined using assays with trypsin (x-axis) and chymotrypsin (y-axis), as in (**a**). The consistency between the two proteases indicates that both proteases are measuring global unfolding, unlike the example in (**a**).

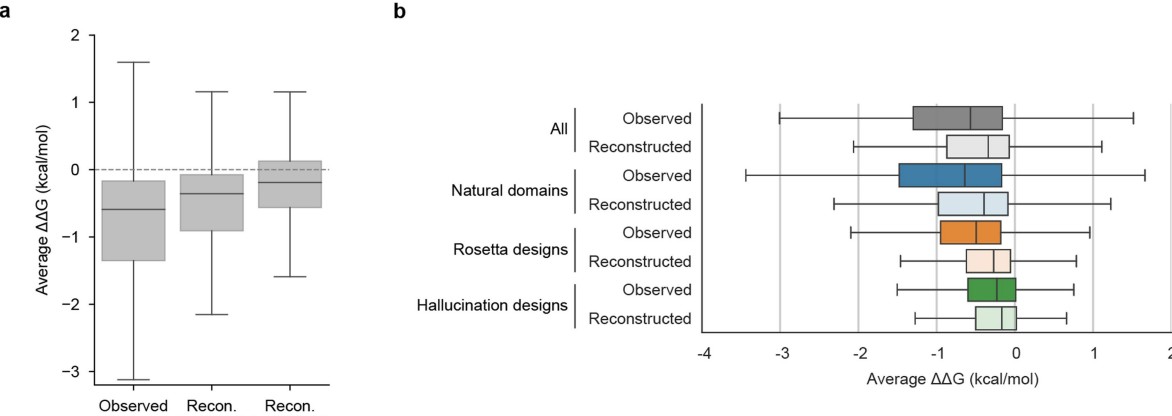

**Extended Data Fig. 7 | Comparison of ΔΔG observed in experiments and ΔΔG reconstructed by the PCs. (a)** Distribution of the mean ΔΔG values across all sites, using the experimental data (left) and ΔΔG values reconstructed from the principal components. More negative average ΔΔG values indicate greater stabilization from the wild-type amino acid. Center line, box limit, and whiskers represent median, upper and lower quartiles, and 1.5x interquartile range of each distribution (n = 17,093). **(b)** As in (**a**), grouped by domain types (natural domains, Rosetta designs, and hallucination designs). Center line, box limit, and whiskers represent median, upper and lower quartiles, and 1.5x interquartile range of each distribution (n = 17,093 in total).

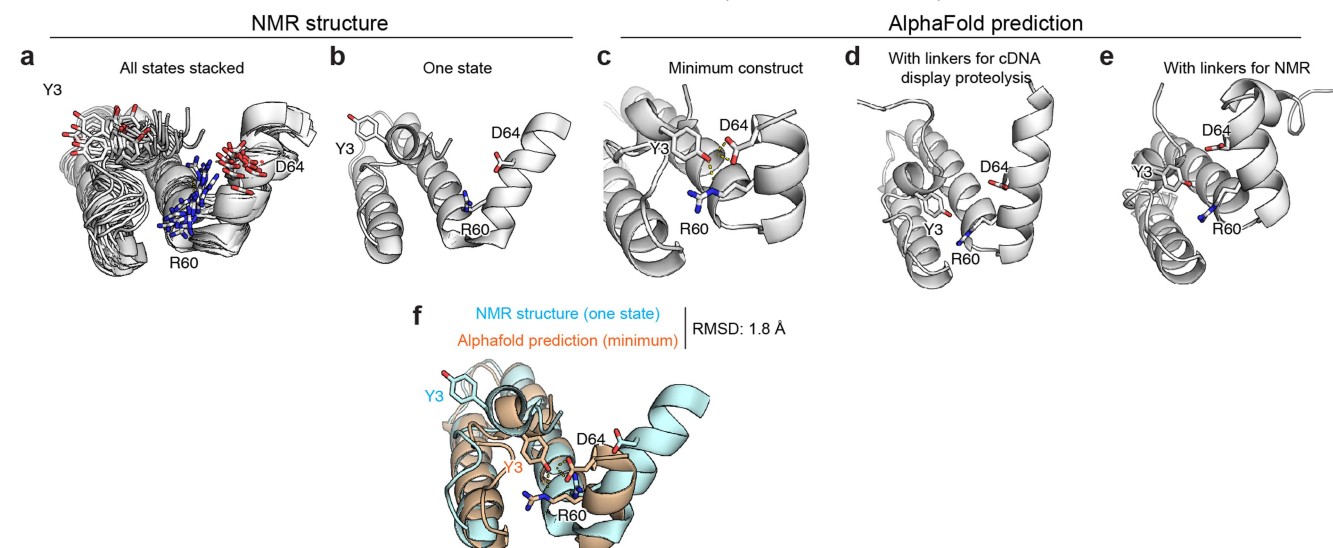

**Extended Data Fig. 8 | Comparison of AlphaFold model and NMR structure for J domain of HSJ1a Structure of J domain in HSJ1a (2LGW).** We show NMR structure of all states stacked (a) and the first state (b), and AlphaFold predicted structures for the minimum construct (the variable segment in cDNA display) (c), the construct with linkers for cDNA display proteolysis (d), and the exact sequence used for NMR (e). In (f), we overlay the first state of the NMR ensemble (cyan) with the AlphaFold structure (orange) of the minimal construct.

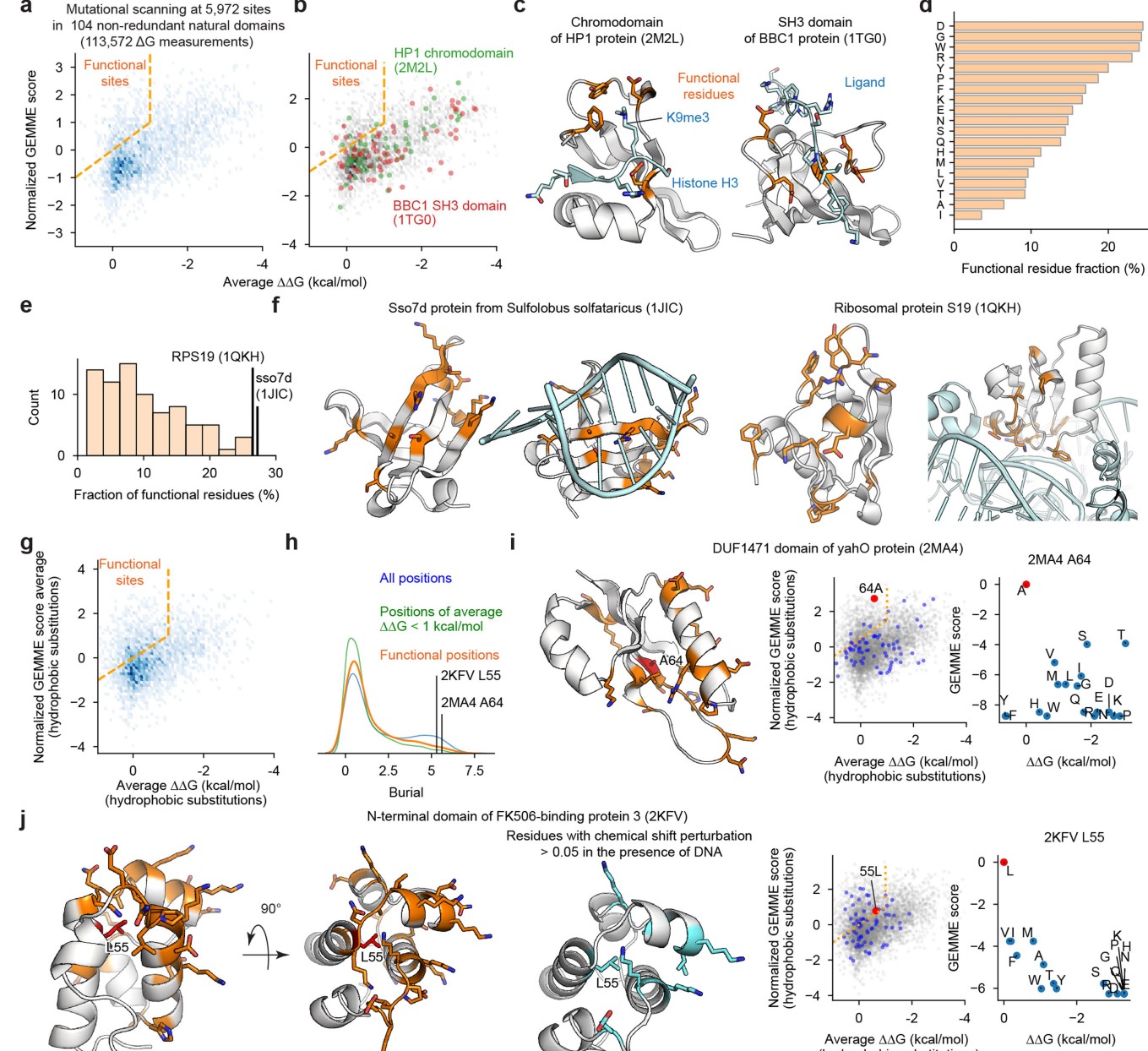

**Extended Data Fig. 9 | Properties of functional sites across diverse domains.** (**a**) The relationship between wild-type stability (average ΔΔG for substitutions) and evolutionary-based sensitivity to substitutions (normalized averaged GEMME score). All sites above the orange dashed line are highly conserved but unimportant for stability; we define these as "functional sites". (**b**) As in (a), highlighting positions in the HP1 chromo domain (2M2L; green) and the BBC1 SH3 domain (1TG0; red). (**c**) Structures of HP1 chromo domain and BBC1 SH3 domain (gray) and their ligands (light blue). Functional sites are shown in orange. Ligand positions were modeled based on PDB structures 1KNA (for HP1) and 2LCS (for the SH3 domain). (**d**) Amino acids are ranked by the percentage of positions where that wild-type amino acid is classified as functional, for positions in 104 non-redundant natural domains. (**e**) The percentage of functional residues in each of the 104 non-redundant domains. (**f**) Structures of the two domains with the highest percentages of functional residues. Nucleic acids interacting with each of the structures are shown in light blue and functional residues are shown in orange. The Sso7d-DNA complex is the crystal structure 1BNZ; the S19-RNA complex is modeled based on the 4V5Y structure. (**g**) As in (a), except only considering nonpolar

substitutions for calculating ΔΔG and normalized averaged GEMME score. (**h**) The distributions of burial (side chain contacts) for all sites (blue), sites where the wild-type amino acid is unimportant for stability (average ΔΔG < 1 kcal/mol) (green), and functional sites (orange). Functional sites are generally located on the surface of the protein. Two unusual buried functional residues are highlighted. (**i**) Structure of the DUF1471 domain of yahO (2MA4) with functional sites in orange and the unusual buried functional site A64 in red. Ala64 is highly conserved yet the domain is stabilized by substitutions to Tyr or Phe (positive ΔΔG, x-axis). However, Tyr and Phe are rarely found in evolution (low GEMME score, y-axis). (**j**) Left: Structure of the N-terminal domain of FK506-binding protein 3 (2KFV) with functional sites in orange and the unusual buried functional site L55 (L78 in PDB numbering) in red. Middle: Residues with chemical shift perturbations in response to DNA binding[43]; L55 shows a perturbation despite not contacting DNA. Right: L55 is conserved (high GEMME score, y-axis) but relatively unimportant for stability (low average ΔΔG, x-axis). Substitution to Phe, Val, or Ile is thermodynamically neutral (ΔΔG near zero) but these amino acids are rarely found in evolution (low GEMME score).

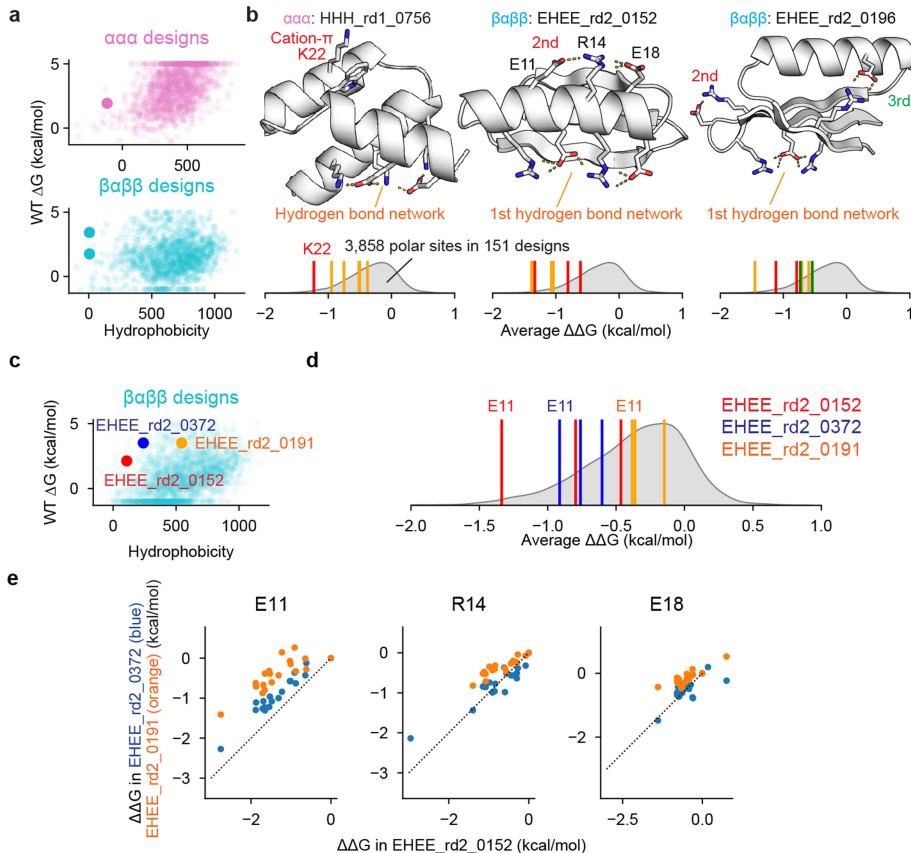

**Extended Data Fig. 10 | Analysis of stable yet less hydrophobic designs and notable hydrogen bond networks.** (**a**) Relationship between hydrophobicity (calculated based on the previous report[86]) and folding stability (ΔG) for designed proteins[18]. Examples from (b) are shown as large dots. (**b**) For three proteins with high folding stability and low hydrophobicity, we highlight critical hydrophilic interactions stabilizing these proteins. Gray density plots show the average ΔΔG of substitutions at 3,858 polar sites in 151 designed domains. Colored vertical bars show the values for the highlighted positions. These three proteins feature polar amino acids where the average ΔΔG of substitutions is unusually destabilizing (> top 5%ile). For HHH_rd1_0756, K22 is shown as a red line; the interacting W32 is considered nonpolar and not shown. Full mutational scanning results are shown in Supplementary Fig. 7. All three structures are design models reported previously[18], not experimental structures. (**c**) As in (**a**), highlighting EHEE_rd2_0152 (from (**b**)) and two other

designs with the same hydrogen bond network. (**d**) Average ΔΔG of substitutions at 3,715 polar sites in 144 designed domains. The colored vertical bars indicate the values for the sites related to the 2nd hydrogen bond network shown in (**b**) for EHEE_rd2_0152. (**e**) Relationship between ΔΔG in EHEE_rd2_0152 and in the other designs EHEE_rd2_0372 or EHEE_rd2_0191 for E11, R14, and E18. At E11, substitutions to the 19 other amino acids have smaller effects in EHEE_rd2_0372 (blue) and EHEE_rd2_0191 (orange) compared to in EHEE_rd2_0152 (e.g. all points are above the dashed Y=X line). However, the points are ordered similarly; i.e. the rank ordering of the 19 other amino acid variants in stability is similar between the three designs. For R14 and E18, substitutions in EHEE_rd2_372 (blue) have similar effect sizes to EHEE_rd2_0152, but substitutions in EHEE_rd2_0191 (orange) have smaller effects. Again, the rank ordering of the amino acid variants by stability is similar across the three designs.

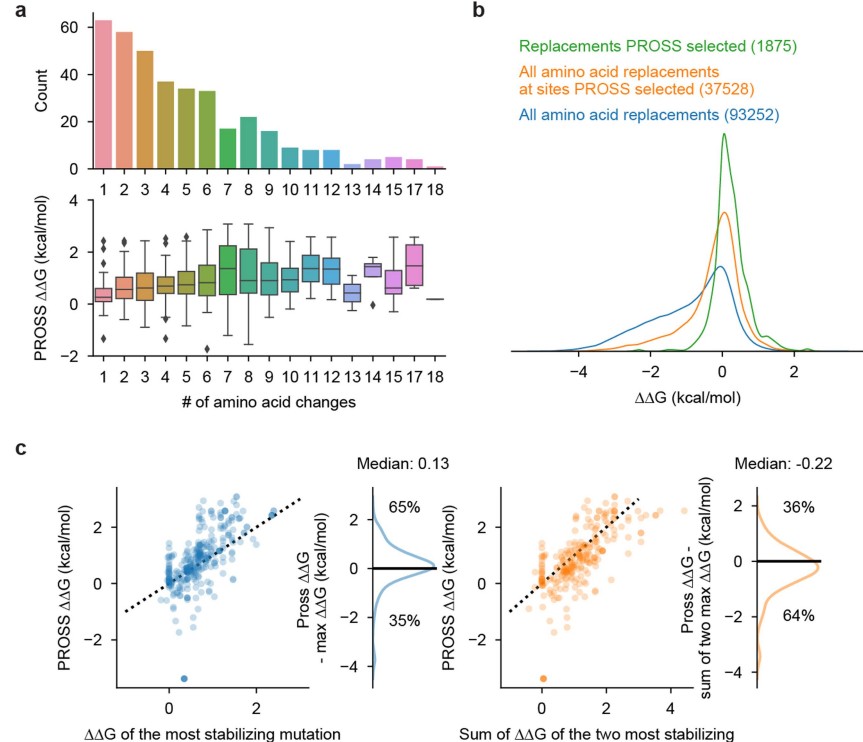

**Extended Data Fig. 11 | Global analysis of PROSS designs. (a)** All 727 PROSS designs grouped according to the number of amino acid substitutions in each design. Top: the number of designs with each different number of substitutions. Bottom: the distribution of design results for each group. ΔΔG indicates the stability of the PROSS design (ΔG) minus the stability of the original wild-type sequence; positive ΔΔG indicates the design stabilized the domain. Center line, box limit, whiskers, and dots represent median, upper and lower quartiles, 1.5x interquartile range, and outliers of each distribution (n = 727 in total). **(b)** ΔΔG distributions for all amino acid substitutions in wild-type domains used as input to PROSS (blue), all amino acid substitutions at sites modified in PROSS designs (orange), and all PROSS-designed substitutions (green). All ΔΔG measurements are in the original wild-type background; positive ΔΔG indicates

stabilizing substitutions. **(c)** Relationship between ΔΔG of PROSS designs and ΔΔG of the most stabilizing mutations designed by PROSS. At left, we compare PROSS designs to the single most stabilizing mutation (in the original wild-type background) out of all the substitutions in the PROSS design. At right, we compare PROSS designs to the sum of the two most stabilizing mutations (each measured individually in the original wild-type background without considering thermodynamic coupling). The density plots show the distribution of PROSS designs that were better (positive) or worse (negative) than the single best mutation (left) or sum of the two best mutations (right). Two-thirds of designs are better than the best single designed mutation, although the difference is small. Likewise, two-thirds of designs are worse than the additive effect of the two best designed mutations (assuming no thermodynamic coupling).

**Extended Data Table 1 | Number of sequences and sequence groups for our datasets**

| Dataset name | Description | # of total sequences | Sequence group | # of sequence groups | # of sequences |
|---|---|---|---|---|---|
| Dataset #1 | All data (no filtering) | 1,841,285 | Single a.a. mut. | 983 wild-types | 1,046,752 |
| | | | Double + triple a.a. mut. | 725 pairs (including 36 triples) | 416,274 |
| | | | Scrambles for unfolded model | - | 68,427 |
| | | | Rocklin 2017 rd1-3 | - | 36,707 |
| | | | Others | - | 273,125 |
| Dataset #2 | All data for ΔG | 776,298 | Single a.a. mut. | 478 wild-types | 566,180 |
| | | | Double a.a. mut. | 559 pairs | 210,118 |
| Dataset #3 | All data for ΔΔG (WT < 4.75 kcal/mol to detect stabilizing mutations) | 607,839 | Single a.a. mut. | 412 wild-types | 448,788 |
| | | | Double a.a. mut. | 496 pairs | 159,051 |
| Dataset #4 (For Fig. 3) | Dataset #3 after removing modified/variant WTs | 325,132 | Single a.a. mut. | 365 wild-types | 325,132 |
| Dataset #5 (For Fig. 5) | Non-redundant natural domains (for amino acid classification model) | 113,572 | Single a.a. mut. | 104 wild-types | 113,572 |

# Reporting Summary

## Statistics

For all statistical analyses, confirm that the following items are present in the figure legend, table legend, main text, or Methods section.

| n/a | Confirmed | |
|---|---|---|
| ☐ | ☒ | The exact sample size ($n$) for each experimental group/condition, given as a discrete number and unit of measurement |
| ☐ | ☒ | A statement on whether measurements were taken from distinct samples or whether the same sample was measured repeatedly |
| ☒ | ☐ | The statistical test(s) used AND whether they are one- or two-sided *Only common tests should be described solely by name; describe more complex techniques in the Methods section.* |
| ☒ | ☐ | A description of all covariates tested |
| ☒ | ☐ | A description of any assumptions or corrections, such as tests of normality and adjustment for multiple comparisons |
| ☐ | ☒ | A full description of the statistical parameters including central tendency (e.g. means) or other basic estimates (e.g. regression coefficient) AND variation (e.g. standard deviation) or associated estimates of uncertainty (e.g. confidence intervals) |
| ☐ | ☒ | For null hypothesis testing, the test statistic (e.g. $F$, $t$, $r$) with confidence intervals, effect sizes, degrees of freedom and $P$ value noted *Give P values as exact values whenever suitable.* |
| ☐ | ☒ | For Bayesian analysis, information on the choice of priors and Markov chain Monte Carlo settings |
| ☒ | ☐ | For hierarchical and complex designs, identification of the appropriate level for tests and full reporting of outcomes |
| ☐ | ☒ | Estimates of effect sizes (e.g. Cohen's $d$, Pearson's $r$), indicating how they were calculated |

*Our web collection on statistics for biologists contains articles on many of the points above.*

## Software and code

Policy information about availability of computer code

| Data collection | Rosetta Macromolecular Modeling Suit 3.10; TrRosetta hallucination (e3e408f); CFX Maestro Software v2.3 |
|---|---|
| Data analysis | Python 3.9; DNAWorks2.0; https://github.com/Rocklin-Lab/cdna-display-proteolysis-pipeline |

For manuscripts utilizing custom algorithms or software that are central to the research but not yet described in published literature, software must be made available to editors and reviewers. We strongly encourage code deposition in a community repository (e.g. GitHub). See the Nature Portfolio guidelines for submitting code & software for further information.

## Data

Policy information about availability of data

All manuscripts must include a data availability statement. This statement should provide the following information, where applicable:
- Accession codes, unique identifiers, or web links for publicly available datasets
- A description of any restrictions on data availability
- For clinical datasets or third party data, please ensure that the statement adheres to our policy

All data and codes are available in the main text, the Extended Data Figs or Tables, or available for download at https://doi.org/10.5281/zenodo.7992926 or https://github.com/Rocklin-Lab/cdna-display-proteolysis-pipeline

# Human research participants

Policy information about studies involving human research participants and Sex and Gender in Research.

| | |
|---|---|
| Reporting on sex and gender | NA |
| Population characteristics | NA |
| Recruitment | NA |
| Ethics oversight | NA |

Note that full information on the approval of the study protocol must also be provided in the manuscript.

# Field-specific reporting

Please select the one below that is the best fit for your research. If you are not sure, read the appropriate sections before making your selection.

☒ Life sciences  ☐ Behavioural & social sciences  ☐ Ecological, evolutionary & environmental sciences

For a reference copy of the document with all sections, see nature.com/documents/nr-reporting-summary-flat.pdf

# Life sciences study design

All studies must disclose on these points even when the disclosure is negative.

| | |
|---|---|
| Sample size | The data is large-scale screening and the data was automatically processed through the pipeline. There is no need to pre-define the sample size. |
| Data exclusions | We excluded some data with low reliability in analysis (95% confidence interval >0.5 for K50, >0.66 for ΔG). We described the detail in captions or method section. |
| Replication | The reproducibility of cDNA display proteolysis was checked in SI Fig2 |
| Randomization | The data is large-scale screening and the data was automatically processed through the pipeline. There is no need to randomize the data. |
| Blinding | The data is large-scale screening and the data was automatically processed through the pipeline. There is no need to blind the data. |

# Reporting for specific materials, systems and methods

We require information from authors about some types of materials, experimental systems and methods used in many studies. Here, indicate whether each material, system or method listed is relevant to your study. If you are not sure if a list item applies to your research, read the appropriate section before selecting a response.

## Materials & experimental systems

| n/a | Involved in the study |
|---|---|
| ☐ | ☒ Antibodies |
| ☒ | ☐ Eukaryotic cell lines |
| ☒ | ☐ Palaeontology and archaeology |
| ☒ | ☐ Animals and other organisms |
| ☒ | ☐ Clinical data |
| ☒ | ☐ Dual use research of concern |

## Methods

| n/a | Involved in the study |
|---|---|
| ☒ | ☐ ChIP-seq |
| ☒ | ☐ Flow cytometry |
| ☒ | ☐ MRI-based neuroimaging |

## Antibodies

| | |
|---|---|
| Antibodies used | PA-tag antibody (NZ-1 FUJIFILM Wako Chemicals) |
| Validation | PA-tag antibody was validated in Fujii et al. Protein Expression and Purification 2014 https://doi.org/10.1016/j.pep.2014.01.009 |

