## [Peer Review File · Nature]

Manuscript Title:

Mega-scale experimental analysis of protein folding stability in biology and design

Editorial Notes:

Redactions – Third Party Material

Reviewer Comments & Author Rebuttals

Reviewer Reports on the Initial Version:

Referee #1:

Computational methods for protein structure prediction and protein design have improved greatly over the last few years as new AI-based methods have been applied to these problems. Critical for this success are the large databases of protein sequences and structures that have allowed training of large neural networks that include millions of adjustable parameters. One problem in protein modeling that has been historically difficult is predicting how a set of amino acid mutations will affect the stability of a protein. This is an important problem for understanding how mutations lead to disease and underlies protein engineering efforts aimed at improving therapeutic proteins by increasing their thermostability. Until now, it has not been possible to directly apply deep learning to this problem because the datasets of experimentally determined protein stabilities are relatively small (on the order of 10,000 protein variants). This study describes an approach that allows the stabilities of millions of protein variants to be measured in just a few weeks. Overall, the quality of the work presented here is high, and several controls are performed to make sure that the reported data is accurate. Importantly, the data correlate well with thermostability data that has been collected using more traditional approaches.

The large number of proteins characterized in this study allow the authors to make several interesting observations. (1) Using principle component analysis they are able to infer which amino acid biophysical properties correlate the most with protein stability and then ask if the most important aa features (hydrophobicity, secondary structure preference, etc ...) can be used to explain the aa preferences at specific sites in designed proteins and naturally-occurring proteins. Their results show that designed proteins are more often stabilized using canonical aa features while naturally-occurring proteins make use of more nuanced/specialized interactions. (2) By comparing the relative stabilities of the WT aa with other aa at each residue in their data set they probe how much naturally-occurring sequences are optimized for protein stability versus other features. The results clearly show that some amino acids (particularly the polar amino acids) are highly represented in protein sequences to provide solubility more than stability. (3) This previous finding leads to the natural conclusion that protein design methods such as Rosetta/ProteinMPNN that are optimized to reproduce native sequences may not be the best approaches for finding mutations that will improve protein stability. This result is highlighted by showing that the web server PROSS (which is based on MSAs and Rosetta) can identify stabilizing mutations but there appears to be significant room for improvement. Given these observations and the strong need for datasets such as this one, the paper is appropriate for publication in Nature and should have a high impact on the field.

Specific issues for consideration during revision:

- Overall, the text and is well written and should be understandable by a none-expert. In contrast, some of the figures are quite technical/complicated, and it may be challenging for a broad audience to understand what is being represented. In particular, I found myself struggling to understand Figs 1b,c, 3d, and 5c. Can some explanation be given for how the x-axis should be interpreted in 3d? What does it mean for PC1 to be positive? Does it mean the amino acids on the

right side of the figure above (Fig. 3c) are more favored? In the fourth panel in 3d, does the curve labeled "basic" indicate contacts made with basic residues? If so, this could be more precisely explained. I think I understand 5c, it just took a while. In the text no comment is made about the steeper slope of the green line below 2 kcal/mol. This seems like an interesting result that makes sense, i.e. stabilizing the protein by 1 kcal/mol is more important when the protein is only marginally stable. Once the protein has high stability, there is not much added benefit to adding more stability.

- Is there a way to show in a figure how you get K50 from mostly raw data? I found myself wanting to see a figure that had percent cleaved on the y-axis, [E] on the x-axis, and K50 pointing to the midpoint of the curve. Is it possible to make this plot for representative variants, perhaps the ones shown in Fig. 1d?

- In Fig. 1e the y-axis label "replicate 2" looks like it is part of Fig. 1d. The same is true for the y-axis label in Fig. 1f.

- dG should have the standard state symbol throughout the paper.

Referee #2:

Although the authors have a great deal of data, I'm not sure that we get a lot of new knowledge, and if I am wrong, the limitations on the length of a publication make it impossible to address all this prior knowledge in one paper.

The manuscript represents at least three papers—one on the technique, one on mutagenesis at a combination of locations, and one on improving designed proteins.

I've studied protein chemistry and biophysics for decades, but I could not understand the results section because it is so densely written. I understand the discussion, but without understanding the results, I can't be sure of the veracity of the discussion.

It would have been helpful to discuss the results in terms of ...

**The neutral mutation hypothesis.

e.g., Kimura M. 1985. The role of compensatory neutral mutations in molecular evolution. *Journal of Genetics*, 64: 7-19.

**Various aspects of side chain physical chemistry

e.g., Radzicka A, Wolfenden R. 1988. Comparing the polarities of the amino acids: Side-chain distribution coefficients between the vapor phase, cyclohexane, 1-octanol, and neutral aqueous solution. *Biochemistry*, 27: 1664-1670.

**The mutation matrix

e.g., Gonnet GH, Cohen MA, Benner SA. 1992. Exhaustive matching of the entire protein sequence database. *Science*, 265: 1443-1445.

**Thermodynamic interactions

e.g., any number of papers by Alan Fersht starting with Carter PJ, Winter G, Wilkinson AJ, Fersht AR. 1984. The use of double mutants to detect structural changes in the active site of the tyrosyl-tRNA synthetase (*Bacillus stearothermophilus*). *Cell*, 38: 835-840.

Holder JB, Bennett AF, Chen J, Spencer DS, Byrne MP, Stites WE. 2001. Energetics of side chain packing in staphylococcal nuclease assessed by exchange of valines, isoleucines, and leucines. *Biochemistry*, 40: 13998-14003.

****Other large scale protein chemistry efforts**

e.g., Markin CJ, Mokhtari DA, Sunden F, Appel MJ, Akiva E, Longwell SA, Sabatti C, Herschlag D, Fordyce PM. 2021. Revealing enzyme functional architecture via high-throughput microfluidic enzyme kinetics. *Science*, 373: eabf8761.

Lahr SJ, Broadwater A, Carter CW, Jr., Collier ML, Hensley L, Waldner JL, Pielak GJ, Edgell MH. 1999. Patterned library analysis: A method for the quantitative assessment of hypotheses concerning the determinants of protein structure. *Proceedings of the National Academy of Sciences U S A*, 96: 14860-14865.

To P, Xia Y, Lee SO, Devlin T, Fleming KG, Fried SD. 2022. A proteome-wide map of chaperone-assisted protein refolding in a cytosol-like milieu. *Proceedings of the National Academy of Sciences*, 119: e2210536119.

Two fundamental problems are easily fixed.

The free energy changes as written are zero. The authors mean the standard state free energy change which has a superscript ^o (and a superscript ['] because they made their measurements at pH 7 not pH 0 as they would be in the standard state). The temperature must also be stated.

Significant figures: There is no reason to give more than one significant figure in an uncertainty. For example, 1.76 +/- 1.30 is 2 +/- 1.

Referee #3:

A. Summary of Key Results

Tsuboyama et al. describe cDNA Display Proteolysis for measuring thermodynamic folding stability of hundreds of thousands of protein domains in parallel. This is an elegant extension of cDNA Display technology originally developed by Yamaguchi et al. (2009) for other purposes. The authors estimate folding stabilities for 983? natural and designed small protein domains of 40 – 72 residues. This remarkable data set is analyzed to provide analyses of the impact on protein stability of single-site mutations to all amino acids (as well as single-residue deletions and insertions) at every site in these domains; data on thermodynamic couplings between 595 residue pairs (222,265 double mutants); an analysis of the divergence between sequence variations due to the balance between stability and function; and an exploration of the impact of these data in guiding de novo protein design. Particularly exciting are the large numbers of comprehensive mutational scanning heat maps of effects of mutations on domain stabilities. The authors have also provided extensive data in public-domain GitHub and Zenodo sites, which will be tremendously valuable to the molecular biophysics and structural biology scientific community.

B. Originality and significance

Overall this is a high-quality and high-impact study providing original and new insights in the role of amino acid substitutions on global protein stability for a large number of small, but important, protein domains. The cDNA proteolysis method and its application for htp proteomic stability studies appears to be original to this work. It is very interesting that large numbers of stabilizing mutations were identified – these data suggest an opportunity to better understand how to design stabilizing mutations for enzyme and protein engineering. The cDNA proteolysis method will be very useful for de novo protein design studies.

C. Data and Methodology

My principal concerns involve the question of whether the correlation between proteolytic

susceptibility and thermodynamic stability is consistent across many different domain architectures and specific family members. Although there is a long history of using proteolysis to monitor protein disorder for identification of unfolded states, including early (uncited) work by Scheraga, Saeur, and others, and more recent studies by Rocklin, Baker, and co-workers, there are many potential pitfalls in this analysis. Since the cDNA Display Proteolysis method does not distinguish a single proteolytic cleavage from multiple-extensive hydrolysis of the protein, even local, partial unfolding may be interpreted as "global unfolding". While validating data for 10 small domains is presented in Fig. 2g, even for these well-behaved domains, which may be well-described by 2-state folding/unfolding behavior, there are significant offsets between the ΔG values previously reported and those measured by cDNA Display Proteolysis (Ext. Data Figs 3 and 4); presumably this reflects how different biophysical methods define an "unfolded state". The authors need to better address/discuss these concerns, and better summarize the issues that may impact the accuracy of their thermodynamic unfolding free energy estimates.

Aside from these concerns about inaccuracies in free energy estimates, more discussion is needed about the limitations in terms of the kinds of domains that can be studied with this approach. I suspect the method can be inaccurate for domains with internal disordered loops that can be proteolytically-hydrolyzed, or multidomain proteins with flexible interdomain linkers. It is not clear how many small, natural domains (either studied here or excluded here) are not suitable for this approach because of rapid proteolysis within the domain. Were the domains studied here vetted for suitability or were all small (40 - 70 residue) domains selected as implied in the text? How were disordered N- and C-terminal regions of the domains addressed? What happens to puromycin-protein coupling if the C-terminal residue of the domain is buried? Can mutations impact this coupling efficiency? Does the method work for obligate homodimers? How about disulfide-bonded protein domains (which are not discussed at all), or domains with metal centers that could hold the protein together even after proteolysis? Currently the gene pool synthesis technologies are limited to ~200 bases, creating a different kind of limitation. These several limitations need to be discussed.

Several additional technical details should be addressed or clarified before this work is accepted for publication.

1. I do not understand the data shown in Fig. 1d. Presumably, [F30P]-Protein G is fully unfolded - so the cleavage by trypsin or chymotrypsin is effective even at low enzyme concentrations. But for wt Protein G there seems to be less cleavage with increasing enzyme concentration, but then nearly complete cleavage at high chymotrypsin (but not at high trypsin) concentrations. These data should be explained better.

2. The authors state "We assume that cleavage in the folded state exclusively occurs outside the folded domain (e.g. in the N-terminal PA tag added to all sequences)." Is PA in the B domain of Protein A (which is quite stable)? Why do the authors expect the folded PA to be cleaved by proteases preferentially to the folded domains they include in their libraries?

3. The authors state "We also noticed several variants of Protein GB1 appear unstable in our data but stable in the previous experiments (52). Our structural analysis of these mutations suggests that our measurements are more likely to be correct (Extended Data Fig. 4)". Reviewing Ext. Data Fig. 4, the criticism of these data is handwaving. It is not uncommon that one mutant is well expressed and a similar mutant is not expressed; this does not necessarily discredit the data obtained for the produced mutant. To make this broad conclusion, one would need to attempt to produce the ~16 specific mutants of GB1 exhibiting inconsistent results, assess expression levels, and measure their stability by chemical denaturation (or another method). The authors also point out: "Based on our kinetic model, stability (ΔG) will be underestimated if significant cleavage occurs inside the test domain in the folded state"; could this be an explanation for the discrepancies for these mutants of GB1?

4. "In four domains, mutational scanning revealed trypsin-sensitive loops that could be cleaved in the folded state, leading to inconsistent stabilities between trypsin and chymotrypsin (Extended Data Fig. 7).": However, if the loops in the folded state are BOTH trypsin- and chymotrypsin-sensitive, the results would be correlated, but the interpretation would be an underestimate of the overall fold stability. How often might this occur in these domains?

5. Regarding the NMR vs AF2 models of the J domain of human HSJ1a (shown in Ext. Data Fig. 14). The authors state "However, in the deposited NMR ensemble for 2LGW, the backbone near Y5 (Y3 in our numbering) always positions that residue away from the helix containing R62 and D66, making the interaction network impossible." Panel a shows that the atomic positions in the N-terminal several residues, including residue Y3, are not well-defined (i.e., not reliable) from the NMR data; the N-terminal segment including Y3 appears to be flexible under the NMR conditions. Residue 58, near this network, is a His (in sequence KHK), which could change degree of protonation between the cDNA proteolysis (pH 7.4; PBS) and NMR conditions (pH 6.5). So the difference between the cDNA proteolysis and the NMR data may reflect a pH-dependent partial unfolding of the C-terminal region. One also wonders what is the reliability of the AF2 model in this N-terminal region of the structure (e.g. pLDDT score); the text suggests multiple conformations are predicted. Have the authors selected one conformer from the AF2 output to best support their cDNA proteolysis data, or is there an objective reason to select this particular AF2 conformation?

6. Page 6 refers to 983 natural and designed domains, but the abstract and p. 7 refer to 251 natural and 145 designs, and then on p. 7, 354 natural and 188 de novo designed protein domains. How many domains were studied?

7. Unlike the natural protein domains, I understand that the de novo Rosetta and trRosetta hallucinated designs are theoretical proteins, validated perhaps by AF2 modeling but not by experimental structure determination. This point should be stated in defining these de novo designed domains.

D. Statistical analyses

I have no concerns with the reliability of the statistical analyses.

E. Conclusions

Conclusions are well supported by the data, except for my concerns about the accuracy of the estimates of protein stability based on cDNA proteolysis method, as detailed above.

F. Suggested improvements

Most of the improvements involve revisions to address concerns raised. Experimental studies of the stabilities of ~16 GB1 mutants mentioned in point 3 above would support the validity of the cDNA proteolysis method for estimating thermodynamic stabilities of mutant proteins.

G. Possible additional citations (historic context)

For the early examples of proteolysis for measuring protein stability:

Rupley, J.A.; Scheraga, H.A. *Biochemistry* 1963, 2, 421.

Ooi, T.; Rupley, J.A.; Scheraga, H.A. *Biochemistry* 1963, 2, 432.

Burgess, A.W. et al. *Biochemistry* 1975, 14, 197-200.

Montelione G.T.; Scheraga, H.A. *Acc. Chem. Res.*, 1989 22, 70-76.

Parsell, D.A.; Sauer R.T. *J. Biol. Chem.* 1989 264: 7590-7595.

H. Clarity and context

The paper is very well written.

Gaetano T. Montelione, Ph.D.
Constellation Endowed Chair in Structural Bioinformatics
Professor of Chemistry and Chemical Biology
Rensselaer Polytechnic Institute
<https://montelionelab.chem.rpi.edu>

Author Rebuttals to Initial Comments:

We are grateful for all comments from the three Referees. Below, we describe how we revised the manuscript based on each individual comment. Here is the summary of the revision:

- We revised Fig. 1 to better show how to infer folding stability values from NGS counts (based on Referee #1 comment)
- We revise Fig. 3 to better explain the meaning/interpretations of the PCs (based on Referee #1 comment). This also included removing the previous panels relating to reconstructing the stability landscapes at individual sites. Instead, we replaced these panels with some new (hopefully easier to understand) panels to help explain the PCs.
- We substantially improved our pipeline for filtering good data to ensure data integrity. This improved things on many levels, but especially focused on identifying and removing cases where sequences could be cleaved without global unfolding. We also made the potential pitfalls of the method much more explicit in the main text (based on Referee #3 comment).
- We added new references to important literature (based on Referee #2 and #3 comment)

To help the Referees identify the section we revised, we revised the main text in a different color.

Referee #1:

Computational methods for protein structure prediction and protein design have improved greatly over the last few years as new AI-based methods have been applied to these problems. Critical for this success are the large databases of protein sequences and structures that have allowed training of large neural networks that include millions of adjustable parameters. One problem in protein modeling that has been historically difficult is predicting how a set of amino acid mutations will affect the stability of a protein. This is an important problem for understanding how mutations lead to disease and underlies protein engineering efforts aimed at improving therapeutic proteins by increasing their thermostability. Until now, it has not been possible to directly apply deep learning to this problem because the datasets of experimentally determined protein stabilities are relatively small (on the order of 10,000 protein variants). This study describes an approach that allows the stabilities of millions of protein variants to be measured in just a few weeks. Overall, the quality of the work presented here is high, and several controls are performed to make sure that the reported data is accurate. Importantly, the data correlate well with thermostability data that has been collected using more traditional approaches.

The large number of proteins characterized in this study allow the authors to make several interesting observations. (1) Using principle component analysis they are able to infer which amino acid biophysical properties correlate the most with protein stability and then ask if the most important aa features (hydrophobicity, secondary structure preference, etc ...) can be used to explain the aa preferences at specific sites in designed proteins and naturally-occurring proteins. Their results show that designed proteins are more often stabilized using canonical aa features while naturally-occurring proteins make use of more nuanced/specialized interactions.

(2) By comparing the relative stabilities of the WT aa with other aa at each residue in their data set they probe how much naturally-occurring sequences are optimized for protein stability versus other features. The results clearly show that some amino acids (particularly the polar amino acids) are highly represented in protein sequences to provide solubility more than stability. (3) This previous finding leads to the natural conclusion that protein design methods such as Rosetta/ProteinMPNN that are optimized to reproduce native sequences may not be the best approaches for finding mutations that will improve protein stability. This result is highlighted by showing that the web server PROSS (which is based on MSAs and Rosetta) can identify stabilizing mutations but there appears to be significant room for improvement. Given these observations and the strong need for datasets such as this one, the paper is appropriate for publication in Nature and should have a high impact on the field.

We thank the Referee for their positive evaluation of our study.

Specific issues for consideration during revision:

- Overall, the text and is well written and should be understandable by a none-expert. In contrast, some of the figures are quite technical/complicated, and it may be challenging for a broad audience to understand what is being represented. In particular, I found myself struggling to understand Figs 1b,c, 3d, and 5c. Can some explanation be given for how the x-axis should be interpreted in 3d? What does it mean for PC1 to be positive? Does it mean the amino acids on the right side of the figure above (Fig. 3c) are more favored? In the fourth panel in 3d, does the curve labeled “basic” indicate contacts made with basic residues? If so, this could be more precisely explained.

Thank you for pointing out the need to clarify these points. We made several changes to better explain these results. First, to make this whole section clearer overall, we added the new Fig. 3f showing the sites in one protein structure colored according to the values of five PCs. We also introduced a side-by-side comparison between raw ΔG data and the first five PCs for each site (Fig. 3g). These new panels should help clarify the relationships between the PCs and the structural environments at each site. We also revised the main text to make it clearer that the PC values indicate whether a site is stabilized or destabilized by particular amino acids:

Old text: Based on the loadings of the different amino acids onto each principal component (Fig. 3C), we interpreted the first four components to reflect amino acid hydrophobicity (PC1; 31% of the total variance explained by this PC), helical probability (PC2; 15%), aliphatic vs. aromatic favorability (PC3; 12%), and positive vs. negative charge (PC4; 7%).

---- (Result section, Page. 10)

New text: Based on the loadings of the different amino acids onto each principal component (Fig. 3d), we interpreted the first four components to reflect **whether a site is stabilized by hydrophobic amino acids (PC1; 31% of the total variance explained by this PC), helix-favoring**

amino acids (PC2; 15%), aliphatic vs. aromatic amino acids (PC3; 12%), and positive vs. negative charges (PC4; 7%).

We also added additional text to clarify the reviewer's specific questions:

Can some explanation be given for how the x-axis should be interpreted in 3d [now fig 3E]? What does it mean for PC1 to be positive? Does it mean the amino acids on the right side of the figure above (Fig. 3c) are more favored?

We clarified this by adding panel G (where positive values on PC1 can be seen to correspond to sites that favor hydrophobic amino acids) and by adding the additional text:

----- (Result section, Page. 11)

Sites with positive principal component values (pink) are stabilized by amino acids with positive loadings on that component (Fig. 3d).

In the fourth panel in 3d, does the curve labeled "basic" indicate contacts made with basic residues?

We clarified this with additional text in the caption:

----- (Fig. 3e caption, Page. 13)

Contacts indicate the contact counts at those sites, counting all possible contacts (PC1 and 5), only contacts to aromatic or aliphatic amino acids (PC3) or only contacts to acidic or basic amino acids (PC4).

I think I understand 5c, it just took a while. In the text no comment is made about the steeper slope of the green line below 2 kcal/mol. This seems like an interesting result that makes sense, i.e. stabilizing the protein by 1 kcal/mol is more important when the protein is only marginally stable. Once the protein has high stability, there is not much added benefit to adding more stability.

Thank you for raising this point. We added a short discussion to the main text:

----- (Result section, Page. 17)

The slope is steeper in the low stability region ($\Delta G < 1$ kcal/mol), indicating that folding stability has a larger impact on amino acid selection. However, our 90-protein test set includes only two wild-type sequences with $\Delta G < 1$ kcal/mol, and this may bias this result.

- Is there a way to show in a figure how you get K_{50} from mostly raw data? I found myself wanting to see a figure that had percent cleaved on the y-axis, $[E]$ on the x-axis, and K_{50} pointing to the midpoint of the curve. Is it possible to make this plot for representative variants, perhaps the ones shown in Fig. 1d?

We agree that the plots the referee mentioned should be more clear. We revised the plots as suggested by the referee so that readers can more easily understand the results:

Fig. 1b/c/d: The plots now show the relative sequencing counts or inferred cleavage for each protease concentration. Additionally, we added vertical lines indicating K_{50} values in Fig. 1d. Note that K_{50} is not EC50, but the point where $k_{\text{obs}} = k_{\text{max}}/2$. Therefore, the K_{50} lines point to $\exp(-k_{\text{max}}/2)$ (~ 0.10). We also added this note in the plot and the caption.

- In Fig. 1e the y-axis label “replicate 2” looks like it is part of Fig. 1d. The same is true for the y-axis label in Fig. 1f.

Related to the point above, we removed the plot. However, based on the Referee’s suggestion, we optimized the space between the axis and labels.

- dG should have the standard state symbol throughout the paper.

As the Referee (and Referee #2) pointed out, we added the explanation of what we meant by “ ΔG ”:

----- (Result section, Page. 5)

“ ΔG°_{298K} for unfolding at pH 7.4 298 K[room temperature]; hereafter, ΔG ”

Referee #2:

Although the authors have a great deal of data, I'm not sure that we get a lot of new knowledge, and if I am wrong, the limitations on the length of a publication make it impossible to address all this prior knowledge in one paper.

The manuscript represents at least three papers—one on the technique, one on mutagenesis at a combination of locations, and one on improving designed proteins.

I've studied protein chemistry and biophysics for decades, but I could not understand the results section because it is so densely written. I understand the discussion, but without understanding the results, I can't be sure of the veracity of the discussion.

We thank the Referee for their critical comments and evaluations of our current manuscript. We apologize that our manuscript is too dense, but with help of all Referees' comments, we made our best effort to make it possible to address all prior knowledge and results in one paper with a reasonable length.

It would have been helpful to discuss the results in terms of ...

**The neutral mutation hypothesis.

e.g., Kimura M. 1985. The role of compensatory neutral mutations in molecular evolution. *Journal of Genetics*, 64: 7-19.

**Various aspects of side chain physical chemistry

e.g., Radzicka A, Wolfenden R. 1988. Comparing the polarities of the amino acids: Side-chain distribution coefficients between the vapor phase, cyclohexane, 1-octanol, and neutral aqueous solution. *Biochemistry*, 27: 1664-1670.

**The mutation matrix

e.g., Gonnet GH, Cohen MA, Benner SA. 1992. Exhaustive matching of the entire protein sequence database. *Science*, 265: 1443-1445.

**Thermodynamic interactions

e.g., any number of papers by Alan Fersht starting with Carter PJ, Winter G, Wilkinson AJ, Fersht AR. 1984. The use of double mutants to detect structural changes in the active site of the tyrosyl-tRNA synthetase (*Bacillus stearothermophilus*). *Cell*, 38: 835-840.

Holder JB, Bennett AF, Chen J, Spencer DS, Byrne MP, Stites WE. 2001. Energetics of side chain packing in staphylococcal nuclease assessed by exchange of valines, isoleucines, and leucines. *Biochemistry*, 40: 13998-14003.

**Other large scale protein chemistry efforts

e.g., Markin CJ, Mokhtari DA, Sunden F, Appel MJ, Akiva E, Longwell SA, Sabatti C, Herschlag D, Fordyce PM. 2021. Revealing enzyme functional architecture via high-throughput microfluidic enzyme kinetics. *Science*, 373: eabf8761.

Lahr SJ, Broadwater A, Carter CW, Jr., Collier ML, Hensley L, Waldner JL, Pielak GJ, Edgell MH. 1999. Patterned library analysis: A method for the quantitative assessment of hypotheses concerning the determinants of protein structure. *Proceedings of the National Academy of Sciences U S A*, 96: 14860-14865.

To P, Xia Y, Lee SO, Devlin T, Fleming KG, Fried SD. 2022. A proteome-wide map of chaperone-assisted protein refolding in a cytosol-like milieu. *Proceedings of the National Academy of Sciences*, 119: e2210536119.

We thank the Referee for introducing useful literature we should cite in our manuscript. Although we could not cite all of them or discuss them in depth, we added additional citations to Fersht 1984, Markin 2021, and To 2022.

Two fundamental problems are easily fixed.

The free energy changes as written are zero. The authors mean the standard state free energy change which has a superscript o (and a superscript ' because they made their measurements at pH 7 not pH 0 as they would be in the standard state). The temperature must also be stated.

As the Referee (and Referee #1) pointed out, we added the standard state symbol and temperature to the paper. We write:

----- (Result section, Page. 5)

“ ΔG°_{298K} for unfolding at pH 7.4 298 K[room temperature]; hereafter, ΔG ”

Significant figures: There is no reason to give more than one significant figure in an uncertainty. For example, 1.76 +/- 1.30 is 2 +/- 1.

We agree we do not need the additional significant figures past 1 decimal for folding stability. We fixed all points.

Referee #3:

A. Summary of Key Results

Tsuboyama et al. describe cDNA Display Proteolysis for measuring thermodynamic folding stability of hundreds of thousands of protein domains in parallel. This is an elegant extension of cDNA Display technology originally developed by Yamaguchi et al. (2009) for other purposes. The authors estimate folding stabilities for 983? natural and designed small protein domains of 40 – 72 residues. This remarkable data set is analyzed to provide analyses of the impact on protein stability of single-site mutations to all amino acids (as well as single-residue deletions and insertions) at every site in these domains; data on thermodynamic couplings between 595 residue pairs (222,265 double mutants); an analysis of the divergence between sequence variations due to the balance between stability and function; and an exploration of the impact of these data in guiding de novo protein design. Particularly exciting are the large numbers of comprehensive mutational scanning heat maps of effects of mutations on domain stabilities. The authors have also provided extensive data in public-domain GitHub and Zenodo sites, which will be tremendously valuable to the molecular biophysics and structural biology scientific community.

B. Originality and significance

Overall this is a high-quality and high-impact study providing original and new insights in the role of amino acid substitutions on global protein stability for a large number of small, but important, protein domains. The cDNA proteolysis method and its application for htp proteomic stability studies appears to be original to this work. It is very interesting that large numbers of stabilizing mutations were identified – these data suggest an opportunity to better understand how to design stabilizing mutations for enzyme and protein engineering. The cDNA proteolysis method will be very useful for de novo protein design studies.

We thank the Referee for their detailed evaluation and interest in our work.

C. Data and Methodology

My principal concerns involve the question of whether the correlation between proteolytic susceptibility and thermodynamic stability is consistent across many different domain architectures and specific family members. Although there is a long history of using proteolysis to monitor protein disorder for identification of unfolded states, including early (uncited) work by Scheraga, Saeur, and others, and more recent studies by Rocklin, Baker, and co-workers, there are many potential pitfalls in this analysis. Since the cDNA Display Proteolysis method does not distinguish a single proteolytic cleavage from multiple-extensive hydrolysis of the protein, even local, partial unfolding may be interpreted as “global unfolding”. While validating data for 10 small domains is presented in Fig. 2g, even for these well-behaved domains, which may be well-described by 2-state folding/unfolding behavior, there are significant offsets between the ΔG values previously reported and those measured by cDNA Display Proteolysis (Ext. Data

Figs 3 and 4); presumably this reflects how different biophysical methods define an “unfolded state”. The authors need to better address/discuss these concerns, and better summarize the issues that may impact the accuracy of their thermodynamic unfolding free energy estimates.

We appreciate that the Referee raised the potential pitfalls of our cDNA proteolysis assay. We completely agree with the Referee that our assay only works for proteins that behave in a two-state manner and are cleaved primarily from the globally unfolded state. This is a really important point that the manuscript needed to address more clearly. We made several large and small changes to the manuscript to better explain the limitations of the experiment and to ensure the quality of our data.

First, we revised the text to point out how any cleavage from folded state(s) can distort the assay:

Old text: Based on our kinetic model, (1) stability (ΔG) will be underestimated if significant cleavage occurs inside the test domain in the folded state,

---- (Result section, Page. 6)

New text (results): Inferring a unique $K_{50,U}$ for each sequence accounts for differences in unfolded state susceptibility between sequences, but note that any cleavage from folded or partially-folded states cannot be corrected by this model. ... Based on our kinetic model, (1) stability (ΔG) will be underestimated if significant cleavage occurs inside the test domain from folded or partially folded states (i.e. without global unfolding),

Additionally in discussion:

Old text: Many domains - particularly larger protein structures - will not satisfy these conditions, and issues such as non-cooperativity, kinetic stability, or aggregation are invisible in a single measurement.

---- (Discussion section, Page. 25)

New text: Many domains - particularly larger protein structures - will not satisfy these conditions. Our deep mutational scanning data often suggested cases of non-cooperativity or potential aggregation, but these potential artifacts can be invisible when assaying individual sequences without mutational scanning. Furthermore, cleavage from folded state(s) may be undetectable even with mutational scanning data if both proteases are equally affected and the cleavage is not overly sensitive to any individual mutation.

Second, we substantially revised how we classify mutational scanning data to more stringently remove data from domains that may be cleaved from folded state(s). We identified these domains by searching for wild-type amino acids that are protease cut sites where removal of the cut site leads to apparent stabilization by one protease and not the other. These domains are

now classified as “Group 11: Likely cleaved in folded state(s)”. Collectively, our more stringent filtering (for possible folded state cleavage or other interference) lowered the number of domains classified as “Good” (G0 or G1) from 560 total domains down to 478 total domains. This also lowered the total number of sequences classified as “Good” from 851,552 to 776,298.

In addition to removing entire domains that showed evidence of cleavage from folded or partially-folded states, we also expanded our filtering to remove individual sequences where a newly introduced mutation may cause cleavage from the folded state. We identified these sequences in our mutational scanning data by searching for positions that are insensitive to mutation (i.e. in a likely unstructured region) where adding a new cut site leads to apparent destabilization from one protease but not the other. We describe this filtering in the main text:

---- (Result section, Page. 7; starting in red)

Deep mutational scanning of hundreds of domains revealed several overall patterns. The largest fraction of these domains showed clear, biophysically reasonable sequence-stability relationships that were consistent between independent experiments with trypsin and chymotrypsin. However, other domains were completely unfolded, too stable to resolve, or produced inconsistent results between the proteases (Fig. 2b). To construct a reliable dataset of ΔG and $\Delta\Delta G$ measurements from folded domains, we filtered the mutational scans based on the consistency of the independent trypsin and chymotrypsin ΔG estimates and other criteria designed to identify domains showing excessive cleavage from folded or partially folded states (Extended Data Fig. 7 and see Methods for details on how we filtered entire domains and individual mutants). Many domains that showed evidence of cleavage from folded states had cleavable sites near the termini or in flexible loop regions. Indeed, four of our deep mutational scans employ modified wild-type backgrounds in which folded state cleavage sites (identified in our first mutational scans) have been removed (Extended data Fig. 8). However, other domains with long loops appeared fully cooperative, without evidence of folded state cleavage (Extended Data Fig. 9).

Along with these changes to our data filtering, we substantially expanded our description of this filtering in the Supporting Information (See section “Data quality filtering and classification of Datasets #1, #2, and #3 (For Fig. 2)” Page. 36-38). In that section, we describe how poor correlations, slopes, or intercepts can be signs of folded state cleavage with one protease or the other, and how we removed these data from our analysis. We also describe how we further filtered out domains showing evidence of cleavage from folded state(s) even when these domains passed other criteria. This specific part reads:

---- (Method section, Page. 38)

Group 11: Likely cleavable in folded state(s). In many cases, excessive cleavage from the folded state or partially folded states will lead to low wild-type stability (G3), poor correlation between

the proteases (G5), or a poor slope (G6). However, we saw some evidence of folded state cleavage even in mutational scans that passed these filtering criteria. Specifically, we observed cases where mutating out a wild-type cut site led to increased protease resistance (higher K_{50}) and apparently higher stability (ΔG) to one specific protease but not the other (e.g. R16 in Extended Data Fig. 8a). This increase in apparent stability for just one protease suggests that either (1) the site can be cleaved from folded state(s) for that protease, or (2) removing the cut site is decreasing unfolded state susceptibility ($K_{50,U}$) in a way that is not properly accounted for by our model. Because these conditions lower the reliability of our ΔG estimates, we removed these mutational scans from analysis. The code to perform this filtering is provided (Data_quality_filtering_script.ipynb).

Finally, to address the referee's point about the offsets in Fig. 1g:

While validating data for 10 small domains is presented in Fig. 2g, even for these well-behaved domains, which may be well-described by 2-state folding/unfolding behavior, there are significant offsets between the ΔG values previously reported and those measured by cDNA Display Proteolysis (Ext. Data Figs 3 and 4); presumably this reflects how different biophysical methods define an "unfolded state".

Although we agree these offsets could reflect differences between the "unfolded state" observed by proteolysis and the "unfolded state" observed by other methods, we propose in the paper that these offsets are mainly due to differences in experimental conditions. This is supported by Extended Data Fig. 4 (previously Extended Data Fig. 3), which shows that these offsets are closely correlated with the temperature of the reference biophysical measurements. In other words, folding stability for most proteins at 50°C is lower than folding stability at 25°C, so if our experiment is at 25°C and we compare it to published experiments that were at 50°C, there is going to be an offset.

To improve the clarity of this point, we now show directly on Fig. 1g the temperature that each individual reference dataset was collected at.

Aside from these concerns about inaccuracies in free energy estimates, more discussion is needed about the limitations in terms of the kinds of domains that can be studied with this approach. I suspect the method can be inaccurate for domains with internal disordered loops that can be proteolytically-hydrolyzed, or multidomain proteins with flexible interdomain linkers. It is not clear how many small, natural domains (either studied here or excluded here) are not suitable for this approach because of rapid proteolysis within the domain. Were the domains studied here vetted for suitability or were all small (40 - 70 residue) domains selected as implied in the text? How were disordered N- and C-terminal regions of the domains addressed? What happens to puromycin-protein coupling if the C-terminal residue of the domain is buried? Can mutations impact this coupling efficiency? Does the method work for obligate homodimers? How about disulfide-bonded protein domains (which are not discussed at all), or domains with metal centers that could hold the protein together even after proteolysis? Currently the gene pool

synthesis technologies are limited to ~200 bases, creating a different kind of limitation. These several limitations need to be discussed.

We thank the referee for these comments and agree that we need to better clarify all of these points. We added new text and figures throughout the manuscript and supplement to clarify all of these points.

In detail:

I suspect the method can be inaccurate for domains with internal disordered loops that can be proteolytically-hydrolyzed, or multidomain proteins with flexible interdomain linkers.

This point is related to the discussion above about the possibility of cleavage from the folded or partially folded states. We expect our filtering of individual domains and domain mutants should largely remove cases where cleavage can occur from folded or partially folded states, but we now note that this may not be adequate in the discussion:

---- (Discussion section, Page. 25)

Furthermore, cleavage from folded state(s) may be undetectable even with mutational scanning data if both proteases are equally affected and the cleavage is not overly sensitive to any individual mutation.

Due to our new filtering, interested readers can now look at the proteins classified as “Group 11: Likely cleaved in folded state(s)” to see the cases where we believe our data suggests that cleavage is occurring with one or more proteases from folded state(s). In addition to this, we also examined whether proteins with long loops would predictably show cleavage from the folded state. We found that this is not true. Many proteins with long loops show no evidence of cleavage from folded state(s). Several examples are now shown in the new Extended Data Fig. 9, such as the example here:

Although this extended loop contains two chymotrypsin-cleavable phenylalanines and no trypsin-cleavable sites, the trypsin and chymotrypsin ΔG estimates are still very consistent and there is no evidence that removing one or the other phenylalanine leads to any chymotrypsin-specific increase in apparent stability. This suggests we are still measuring global folding stability even in a protein with this extended loop. This is very different from the case shown in Extended Data Fig 8, where the data are very clear that the protein can be cleaved from the folded state by trypsin at Arg16.

“It is not clear how many small, natural domains (either studied here or excluded here) are not suitable for this approach because of rapid proteolysis within the domain. Were the domains studied here vetted for suitability or were all small (40 - 70 residue) domains selected as implied in the text? How were disordered N- and C-terminal regions of the domains addressed?”

We added a new section at the very beginning of the Methods section describing how natural domains were chosen for analysis:

---- (Method section, Page. 27)

Selection of natural proteins for mutational scanning

We first collected all monomeric proteins in the PDB in the 30-100 amino acid length range in June 2021. We next excluded structures that had only a single helix, contained other molecules (e.g., proteins, nucleic acids, metals), were annotated to have DNase/RNase activity or protease inhibition activity, included >4 cystines. We then removed redundant sequences (amino acid sequence distance < 2). We then predicted the structures of these PDB sequences using AlphaFold (even though the PDB structures were known), and used the AlphaFold models to trim amino acids from the N- and C-termini that had a low number of contacts with any other residues. Finally, we selected domains with up to 72 amino acids after excluding N- or C-terminal flexible loop(s).

We also revised the main text to specifically note that we removed disordered terminal regions:

---- (Result section, Page. 7)

To minimize potential cleavage from the folded state, we used AlphaFold models of each natural domain to remove unstructured terminal segments from each sequence.

“What happens to puromycin-protein coupling if the C-terminal residue of the domain is buried? Can mutations impact this coupling efficiency? Does the method work for obligate homodimers? How about disulfide-bonded protein domains (which are not discussed at all), or domains with

metal centers that could hold the protein together even after proteolysis? Currently the gene pool synthesis technologies are limited to ~200 bases, creating a different kind of limitation.”

These are all very important points and we thank the referee for raising them. We addressed these points in two overall ways. First, early in the results section, we now include a discussion of many PDB domains that led to poor data in our assay. Second, in the discussion section, we now explicitly mention all of these limitations of the assay.

---- (Result section, Page. 7)

Many domains that showed evidence of cleavage from folded states had cleavable sites near the termini or in flexible loop regions. Indeed, four of our deep mutational scans employ modified wild-type backgrounds in which folded state cleavage sites (identified in our first mutational scans) have been removed (Extended data Fig. 8). However, other domains with long loops appeared fully cooperative, without evidence of folded state cleavage (Extended Data Fig. 9). Along with susceptibility to cleavage from folded or partially-folded states, other domains taken from the PDB failed in our assay for a variety of reasons, including poor cell-free expression (category G2), structures that were likely stabilized by crystal contacts (e.g. 2CUW, 2FGG), missing N-C cyclization incompatible with cDNA display (e.g. 1E68, 2MP8), poor stability at room temperature (e.g. 2C0S, 2LGN), and large numbers of Cysteine residues (category G9). Our final quality-filtered datasets are shown in Table 1.

---- (Discussion section, Page. 25-26)

The cDNA display proteolysis method massively expands the scale of folding stability experiments. Still, the method currently has notable limitations. First, our assay is limited to proteins that express and fold in the cell-free environment and are compatible with cDNA display, including the N-terminal tag and C-terminal nucleic acid conjugate. Due to the very low concentration of each library member, proteins that are stabilized in complexes are likely unsuitable for this assay. Second, because we digest proteins under native conditions, our inferred thermodynamic stabilities are only accurate when (1) folding is fully cooperative (no unfolded segments get cleaved without global unfolding⁸³), (2) folding is at equilibrium during the assay (no kinetic stability or spurious stability due to aggregation), (3) $K_{50,U}$ is accurately inferred (Fig. 1c), (4) cleavage leads to dissociation of the cDNA (minimal crosslinking (e.g. disulfide bonds) that could retain the C-terminus even after proteolysis), and (5) cleavage rates fall within the measurable range of the assay, which currently limits the dynamic range to ~5 kcal/mol (Fig. 1c). Many domains - particularly larger protein structures - will not satisfy these conditions. Our deep mutational scanning data often suggested cases of non-cooperativity or potential aggregation, but these potential artifacts can be invisible when assaying individual sequences without mutational scanning. Furthermore, cleavage from folded state(s) may be undetectable even with mutational scanning data if both proteases are equally affected and the cleavage is not overly sensitive to any individual mutation. Combining cDNA display proteolysis with chemical denaturation (pulse proteolysis³⁸) may overcome some of these obstacles and

enable mega-scale analysis of less cooperative and/or higher stability proteins, while also avoiding the need to infer $K_{50,U}$. **Although we currently investigate domains with up to 80 amino acids (due to the length limitation of DNA oligo pool synthesis)**, advances in DNA synthesis (including methods like DropSynth^{84,85}) will also make it possible to expand cDNA display proteolysis to analyze diverse libraries of larger domains. Lastly, multiplexed measurements and automated data processing always have the potential to introduce inaccuracies, although we worked to exclude unreliable data. For notable individual results, examining the raw data can be helpful, and we included all data and code to regenerate all fits.

Several additional technical details should be addressed or clarified before this work is accepted for publication.

1. I do not understand the data shown in Fig. 1d. Presumably, [F30P]-Protein G is fully unfolded - so the cleavage by trypsin or chymotrypsin is effective even at low enzyme concentrations. But for wt Protein G there seems to be less cleavage with increasing enzyme concentration, but then nearly complete cleavage at high chymotrypsin (but not at high trypsin) concentrations. These data should be explained better.

As the Referee (and Referee#1) suggested, we included a new plot, new Fig. 1d showing inferred survival (1-cleavage), and how to get K_{50} from NGS data. We believe this plot is more straightforward.

The old Fig. 1d (current Fig. 1c) represents sequencing counts after normalization to no protease counts. When these normalized sequencing counts increase, this does not mean that “there seems to be less cleavage with increasing enzyme concentration” - it means that the sequence is being cleaved less *compared to other sequences in the pool*, which causes its relative fraction of the pool to rise (the fractions of all sequences in the pool must add up to 100%). Hopefully, our new plots 1c and 1d are able to more intuitively explain the relationship between sequencing counts and cleavage.

2. The authors state “We assume that cleavage in the folded state exclusively occurs outside the folded domain (e.g. in the N-terminal PA tag added to all sequences).” Is PA in the B domain of Protein A (which is quite stable)? Why do the authors expect the folded PA to be cleaved by proteases preferentially to the folded domains they include in their libraries?

PA-tag is a 14 amino acid segment of human podoplanin, not in the B domain of Protein A. We revised Fig. 1a to show the length of the PA-tag. The PA-tag is almost completely unfolded as in the following figure, which makes it somewhat susceptible to proteolysis. However, the sequence (EGGVAMPGAEDDVV) lacks basic amino acids and aromatic amino acids for chymotrypsin, so cleavage is still slow.

[REDACTED]

Regarding whether PA is cleaved preferentially to the folded domains, this returns to the question of whether our test domains can be cleaved folded or partially folded states. By stating that “We assume that cleavage in the folded state exclusively occurs outside the folded domain (e.g. in the N-terminal PA tag added to all sequences), so we use an identical $K_{50,F}$ for all sequences.”, we are not trying to say that domains are never cleaved from folded/partially folded state(s) - we are saying that we don't include any possible cleavage from folded state(s) in our model. In other words, for the purposes of the model, we assume this cleavage cannot happen, so that the model can calculate K_{50} s and ΔG s. Then, after we have run the inference model, we use our filtering methods to remove data that we suspect is corrupted due to un-modeled cleavage from folded or partially-folded state(s).

To better clarify this in the text, we revised this paragraph in the results:

----- (Result section, Page. 6)

Based on our kinetic model, (1) stability (ΔG) will be underestimated if significant cleavage occurs inside the test domain **from folded or partially folded states (i.e. without global unfolding)**, (2) stability can be over- or under-estimated depending on the accuracy of $K_{50,U}$ (independent measurements with trypsin and chymotrypsin help correct this), and (3) ΔG values become unreliable if K_{50} approaches $K_{50,F}$ or $K_{50,U}$ (Fig. 1e). **Due to these limitations, we developed data quality filters to remove unreliable ΔG estimates (see below). After filtering, the ΔG values inferred by the model were consistent between our independent experiments with trypsin and chymotrypsin ($R = 0.94$; Fig. 1f).**

3. The authors state “We also noticed several variants of Protein GB1 appear unstable in our data but stable in the previous experiments (52). Our structural analysis of these mutations suggests that our measurements are more likely to be correct (Extended Data Fig. 4)”. Reviewing Ext. Data Fig. 4, the criticism of these data is handwaving. It is not uncommon that one mutant is well expressed and a similar mutant is not expressed; this does not necessarily discredit the data obtained for the produced mutant. To make this broad conclusion, one would need to attempt to produce the ~16 specific mutants of GB1 exhibiting inconsistent results, assess expression levels, and measure their stability by chemical denaturation (or another method). The authors also point out: “Based on our kinetic model, stability (ΔG) will be underestimated if significant cleavage occurs inside the test domain in the folded state”; could this be an explanation for the discrepancies for these mutants of GB1?

We thank the Referee and agree that our data cannot conclusively discredit the published data on GB1. We also agree that cleavage from a folded state or a partially folded state is one possible explanation of the discrepancy between our data and the published GB1 data. Since we agree to these points, instead of trying to definitely resolve the stabilities of the GB1 mutants (which is not the main focus of the paper), we revised our text to remove our criticisms of the published GB1 data. We removed the main text remark about our data being more likely to be correct, and we also removed the “handwaving” explanation from the Extended Data figure caption. Instead, we provide a short summary that we believe describes the major differences in the data:

----- (Caption for Extended Data Fig. 3, Page. 59)

In many cases, the chemical denaturation data indicates that polar substitutions into the hydrophobic core are relatively tolerated (e.g. Y3K, L5R, F30D, F30R, Y45D), whereas cDNA display proteolysis indicates that these substitutions are very destabilizing.

4. “In four domains, mutational scanning revealed trypsin-sensitive loops that could be cleaved in the folded state, leading to inconsistent stabilities between trypsin and chymotrypsin (Extended Data Fig. 7).”: However, if the loops in the folded state are BOTH trypsin- and chymotrypsin-sensitive, the results would be correlated, but the interpretation would be an underestimate of the overall fold stability. How often might this occur in these domains?

We thank the referee for raising this point. We agree that if a domain is cleaved from a folded or partially folded state to a similar degree with both proteases, then the correlation between the proteases could remain strong even though the overall stability is underestimated. In the revised manuscript, we addressed this point by (1) more stringently filtering domains to remove data from domains showing evidence of cleavage from folded states, (2) being more explicit about this important caveat in the main text and discussion, and (3) briefly investigating whether extended loops frequently showed evidence of cleavage from folded states.

1, More stringent filtering. As mentioned, even in cases where trypsin and chymotrypsin data are well-correlated, we now explicitly look for evidence of cleavage from folded states and remove these domains as well. We identified these domains by searching for wild-type amino acids that have protease cut sites where removal of the cut site leads to apparent stabilization by one protease and not the other. These domains are classified as “Group 11: Likely cleaved in folded state(s)” and removed from our main analysis. We acknowledge that this filtering cannot

remove all possible cases of cleavage from folded states, particularly if a loop contains multiple cleavage sites for both proteases that are all accessible from the folded state. However, we would expect in these cases that the overall protease resistance of the protein at this point would be quite low and the domain would be classified as “G3: WT ΔG is too low.” This category was also removed from our main analysis.

2, Being more explicit about this caveat in the results and discussion. In the results, we now state that we filtered for more than just consistency between the two proteases, and that we saw evidence of cleavage from folded state(s):

---- (Result section, Page. 7)

To construct a reliable dataset of ΔG and $\Delta\Delta G$ measurements from folded domains, we filtered the mutational scans based on the consistency of the independent trypsin and chymotrypsin ΔG estimates and other criteria designed to identify domains showing excessive cleavage from folded or partially folded states (Extended Data Fig. 7 and see Methods for details on how we filtered entire domains and individual mutants.) Many domains that showed evidence of cleavage from folded states had cleavable sites near the termini or in flexible loop regions. Indeed, four of our deep mutational scans employ modified wild-type backgrounds in which folded state cleavage sites (identified in our first mutational scans) have been removed (Extended data Fig. 8).

We also added this caveat in the discussion:

---- (Discussion section, Page. 25)

Second, because we digest proteins under native conditions, our inferred thermodynamic stabilities are only accurate when (1) folding is fully cooperative (no unfolded segments get cleaved without global unfolding⁸³) ... Many domains - particularly larger protein structures - will not satisfy these conditions. Our deep mutational scanning data often suggested cases of non-cooperativity or potential aggregation, but these potential artifacts can be invisible when assaying individual sequences without mutational scanning. Furthermore, cleavage from folded state(s) may be undetectable even with mutational scanning data if both proteases are equally affected and the cleavage is not overly sensitive to any individual mutation.

3, Investigating whether unstructured loops always implies cleavage from the folded state. It is difficult to directly answer the Referee’s question “How often might this occur in these domains?”. Many of the proteins filtered out in Group 11 (likely folded state cleavage) showed cleavage sites in loop regions of terminal segments. However, other domains have very long loops with cleavage sites yet still appear to be fully cooperative according to our data. As mentioned above, we show examples of these domains in Extended Data Fig. 9 and added this explanation to the main text:

---- (Result section, Page. 7)

Many domains that showed evidence of cleavage from folded states had cleavable sites near the termini or in flexible loop regions. Indeed, four of our deep mutational scans employ modified wild-type backgrounds in which folded state cleavage sites (identified in our first mutational scans) have been removed (Extended data Fig. 8). However, other domains with long loops appeared fully cooperative, without evidence of folded state cleavage (Extended Data Fig. 9).

Overall, we expect that very few domains would have (1) significant cleavage from the folded state (2) for both proteases (preserving the correlation between independent results with each protease) and (3) have this cleavage be distributed over multiple cleavable sites such that mutational scanning does not detect spurious stabilization due to removing a cut site for one protease. This would require multiple cutting sites for each protease that are accessible from the folded or partially-folded state(s), which would likely lead to low protease resistance overall. Still, because we cannot rule this out across our dataset, we think the best way to handle this point is to be explicit about the possibility through the new text we added in the Results and discussion.

5. Regarding the NMR vs AF2 models of the J domain of human HSJ1a (shown in Ext. Data Fig. 14). The authors state “However, in the deposited NMR ensemble for 2LGW, the backbone near Y5 (Y3 in our numbering) always positions that residue away from the helix containing R62 and D66, making the interaction network impossible.” Panel a shows that the atomic positions in the N-terminal several residues, including residue Y3, are not well-defined (i.e., not reliable) from the NMR data; the N-terminal segment including Y3 appears to be flexible under the NMR conditions. Residue 58, near this network, is a His (in sequence KHK), which could change degree of protonation between the cDNA proteolysis (pH 7.4; PBS) and NMR conditions (pH 6.5). So the difference between the cDNA proteolysis and the NMR data may reflect a pH-dependent partial unfolding of the C-terminal region. One also wonders what is the reliability of the AF2 model in this N-terminal region of the structure (e.g. pLDDT score); the text suggests multiple conformations are predicted. Have the authors selected one conformer from the AF2 output to best support their cDNA proteolysis data, or is there an objective reason to select this particular AF2 conformation?

We thank the Referee for their fruitful comments in our discussion. We did not notice the conditional difference between the NMR analysis (pH 6.5) and our cDNA-display proteolysis (pH 7.4), but the difference could explain the discrepancy between the two assays. We added the point in the section:

---- (Result section, Page. 15)

The strong couplings we identify support the AlphaFold model and suggest the deposited ensemble is missing conserved interactions that form in HSJ1a and other J domain proteins, perhaps due to the specific experimental conditions used.

We also changed the wording in the discussion to state that the hydrogen bonding network is “not present” in the deposited NMR structure instead of “missing”; this language is more neutral about whether it would be correct to show the network.

We used the best AlphaFold2 (AF2) model based on the average pLDDT score (rather than the model best for our data). We actually modeled these structures before collecting any cDNA display proteolysis data, so we did not choose the model to support our data.

---- (Result section, Page. 14)

The interaction network shown in Fig. 4g comes from the AlphaFold predicted structure for our wild-type sequence taken from the J domain of human HSJ1a (the highest confidence model out of five predictions).

6. Page 6 refers to 983 natural and designed domains, but the abstract and p. 7 refer to 251 natural and 145 designs, and then on p. 7, 354 natural and 188 de novo designed protein domains. How many domains were studied?

We apologize for any confusion related to the number of domains. We utilized different sets of DMS data for each figure to analyze the data rationally and to use as large data as possible. To avoid any confusion, we summarized the datasets we used for each figure in the following table and added the following table to the manuscript (Table. 1).

Dataset name	Description	# of total sequences	Sequence group	# of sequence groups	# of sequences
Dataset #1	All data (no filtering)	1,841,285	Single a.a. mut.	983 wild-types	1,046,752
			Double + triple a.a. mut.	725 pairs (including 36 triples)	416,274
			Scrambles for unfolded model	-	68,427
			Rocklin 2017 rd1-3	-	36,707
			Others	-	273,125
Dataset #2	All data for ΔG	776,298	Single a.a. mut.	478 wild-types	566,180
			Double a.a. mut.	559 pairs	210,118
Dataset #3	All data for $\Delta\Delta G$ (WT < 4.75 kcal/mol to detect stabilizing mutations)	607,839	Single a.a. mut.	412 wild-types	448,788
			Double a.a. mut.	496 pairs	159,051
Dataset #4 (For Fig. 3)	Dataset #3 after removing modified/variant WTs	325,132	Single a.a. mut.	365 wild-types	325,132

Dataset #5 (For Fig. 5 and Fig. 6)	Non-redundant natural domains (for amino acid classification model)	113,572	Single a.a. mut.	104 wild-types	113,572
--	---	---------	------------------	----------------	---------

7. Unlike the natural protein domains, I understand that the de novo Rosetta and trRosetta hallucinated designs are theoretical proteins, validated perhaps by AF2 modeling but not by experimental structure determination. This point should be stated in defining these de novo designed domains.

As the Referee pointed out, the structures of de novo proteins have not been validated in experiments, and we include the following description.

----- (Result section, Page. 7)

Note that the structures of Rosetta designs and trRosetta designs have not been validated experimentally.

D. Statistical analyses

I have no concerns with the reliability of the statistical analyses.

E. Conclusions

Conclusions are well supported by the data, except for my concerns about the accuracy of the estimates of protein stability based on cDNA proteolysis method, as detailed above.

F. Suggested improvements

Most of the improvements involve revisions to address concerns raised. Experimental studies of the stabilities of ~16 GB1 mutants mentioned in point 3 above would support the validity of the cDNA proteolysis method for estimating thermodynamic stabilities of mutant proteins.

We thank the Referee for carefully checking the validity of our assay. Overall, we completely agree with the Referee about the limitations of the cDNA display proteolysis method. Based on the Referee's feedback, we are now much explicit about these limitations in the results and discussion, especially related to possible cleavage from folded or partially folded states. And, as we mentioned, we removed our assertions about the reliability of the published GB1 data. Because we agree on the assay's limitations and because knowing the stabilities of the GB1 mutants would not add to our understanding of these limitations, we opted not to add these additional experiments.

G. Possible additional citations (historic context)

For the early examples of proteolysis for measuring protein stability:

Rupley, J.A.; Scheraga, H.A. Biochemistry 1963, 2, 421.
Ooi, T.; Rupley, J.A.; Scheraga, H.A. Biochemistry 1963, 2, 432.
Burgess, A.W. et al. Biochemistry 1975, 14, 197-200.
Montelione G.T.; Scheraga, H.A. Acc. Chem. Res., 1989 22, 70-76.
Parsell, D.A.; Sauer R.T. J. Biol. Chem. 1989 264: 7590-7595.

We appreciate the Referee introducing much useful literature we should cite in our manuscript. Although we could not cite all of them due to the limitation of the citation, we added additional citations to Rupley 1963, Ooi 1963, and Parsell 1989.

H. Clarity and context

The paper is very well written.

We thank the Referee for the positive evaluation of our manuscript.

Reviewer Reports on the First Revision:

Referees' comments:

Referee #1:

The authors have done a great job revising figures and text to address my specific concerns (in particular making the figures easier to understand). As detailed in my previous review, the data from this study (and future studies that are likely to use similar approaches) will be very useful for training deep learning models that can predict how a mutation will influence protein stability.

Referee #3:

Thanks to the authors for their careful and comprehensive responses and revisions made to address concerns I raised. The points I have raised are adequately addressed by these revisions.

Gaetano T Montelione

Author Rebuttals to First Revision:

Referees' comments:

Referee #1:

The authors have done a great job revising figures and text to address my specific concerns (in particular making the figures easier to understand). As detailed in my previous review, the data from this study (and future studies that are likely to use similar approaches) will be very useful for training deep learning models that can predict how a mutation will influence protein stability.

We appreciate the Referee's positive evaluation on our study.

Referee #3:

Thanks to the authors for their careful and comprehensive responses and revisions made to address concerns I raised. The points I have raised are adequately addressed by these revisions.

Gaetano T Montelione

We thank the Referee for their constructive comments, which greatly improved our manuscript.